# Tissue-resident macrophage survival depends on mitochondrial function regulated by SerpinB2 in chronic inflammation

How cellular metabolism facilitates tissue-resident macrophage maintenance remains elusive. Here we show that visceral adipose tissue (VAT)-resident macrophages, unlike monocyte-derived macrophages, are enriched with mitochondrial-specific antioxidant enzymes restraining inflammation and promoting VAT homeostasis and insulin sensitivity. Additionally, VAT resident macrophages express high levels of plasminogen activator inhibitor type 2, encoded by SerpinB2, which is involved in the blood coagulation cascade. SerpinB2 promotes adipose resident macrophage survival by regulating mitochondrial oxidative phosphorylation and preventing the release of pro-apoptotic cytochrome c from the mitochondria into the cytoplasm via anti-oxidant glutathione production. Chronic inflammation, such as obesity, diminishes SerpinB2 expression in VAT macrophages in patients and mice, leading to the decline of this macrophage subset. Mechanistically, interferon-γ elevation in diabetes induces Ikaros, a transcriptional suppressor, which binds to the *SerpinB2* promoter and decreases SerpinB2 expression. Congruently, selective depletion of the IFN-γ receptor in myeloid cells or supplementation of macrophage-specific SerpinB2 deficient mice with N-acetylcysteine, a glutathione precursor, restores VAT resident macrophage survival, decreases adipocyte size, and improves glucose tolerance and insulin sensitivity. Our data thus reveal an unexpected function of SerpinB2 in the regulation of mitochondrial function and survival of tissue-resident macrophages.

Tissue-resident macrophages are important in health and disease, including tissue homeostasis[1], organ regeneration[2–4], removal of cellular debris[5], injury[6], and infection[7]. A recent study showed that peritoneal resident macrophages migrate to the heart after myocardial infarction to mitigate cardiac fibrosis[8]. Cardiac resident macrophages support mitochondrial homeostasis[9] and electrical conduction[10]. A novel CX$_3$C chemokine receptor-positive resident synovial macrophages have been reported to provide a protective barrier against inflammatory reactions[11]. Similarly, microglia, which are resident macrophages in the brain, execute important functions such as synapse plasticity[12] and elimination[13], neuronal activity[14], and tissue repair[15]. Moreover, resident dermal macrophages promote the regeneration of local nerves after mechanical injury and are renewed by local proliferation[4]. Adipose macrophages are indispensable in tissue homeostasis, metabolic inflammation, and insulin resistance[16–42]. Triggering receptors expressed on myeloid cells 2$^+$ lipid-associated adipose macrophages have recently been demonstrated to elicit a protective response in obesity[43].

✉e-mail: duttapa@pitt.edu

Visceral adipose tissue (VAT) harbors two ontogenically different macrophages: monocyte-derived short-lived macrophages and hematopoietic progenitor-derived long-term resident macrophages[44]. Inflammation propagated by monocyte-derived macrophages in VAT is causally linked to increased insulin resistance in obesity[39,45]. In contrast, VAT resident macrophages attenuate the local proinflammatory environment in many pathological conditions including diet-induced obesity[46,47]. Contraction of resident macrophage populations has been reported in inflammatory conditions such as myocardial infarction[44] and infection[48] due to largely unexplored reasons.

Plasminogen activator inhibitors (PAI) are serine protease inhibitors and inactivate tissue and urokinase plasminogen activators, which promote fibrinolysis[49]. Hence PAI increase the risk of thrombosis. PAI-1 is expressed by various cell types including fibroblasts, endothelial cells, adipocytes, cardiomyocytes, and hepatocytes[50]. The plasma concentrations of PAI-1 increase in different pathological conditions such as metabolic and ischemic diseases[51,52]. Genetic and pharmacological inhibitions of PAI-1 ameliorated insulin resistance in mice fed with a high-fat diet (HFD)[53,54]. Moreover, PAI-1 mediates fibrosis of various organs including the heart[55], lungs[56], and kidneys[57], and increases the pathogenesis of coronary artery disease[58], atherosclerosis[59], diabetes[54], and cancer[60,61]. In contrast, to the well-known functions of PAI-1, the physiological roles of PAI-2, which is encoded by *SerpinB2*, are understudied. PAI-2 is inefficiently secreted, and the majority of the protein is retained intracellularly[49]. Unlike PAI-1, PAI-2 is not widely expressed. It is secreted by the placenta and thus, can be readily detected in the plasma during pregnancy. Macrophages are the major cell type that expresses SerpinB2. Yet, the role of SerpinB2 in cellular metabolism in macrophages is largely unknown.

Dysregulated oxidative damage and altered respiratory potential of mitochondria due to impaired antioxidant status are prevalent in apoptotic cells. Among the antioxidant defense systems, glutathione is a key endogenous antioxidant, which acts as a cofactor of many enzymes controlling a multitude of cellular processes, including cell proliferation, growth, and differentiation[62–65]. GSH deficiency is implicated in the etiology of various human diseases including cardiovascular, aging, and diabetes[63,66–71]. However, the effects of GSH on VAT resident macrophage survival and associated inflammation in obesity have not been studied well.

Here we show that inflammatory conditions, such as obesity, diminish SerpinB2 expression in tissue-resident macrophages via the IFN-γ pathway, which results in higher oxidative phosphorylation, lower GSH, and exaggerated release of pro-apoptotic cytoplasmic cytochrome c from the mitochondria into the cytoplasm, leading to macrophage apoptosis. Moreover, selective depletion of the VAT resident macrophage subset enhanced glucose intolerance and insulin resistance in obese mice, accentuating the significance of this macrophage subset in promoting insulin sensitivity. Finally, myeloid-specific *Ifngr1* deficiency and GSH supplementation in *LysM*^cre/+ *SerpinB2*^fl/fl mice curtailed VAT resident macrophage apoptosis, reduced inflammation, and improved glucose tolerance and insulin sensitivity. As such, these findings have therapeutic implications in diseases, such as myocardial infarction and Alzheimer's disease, where resident macrophage functions are altered.

## Results

### VAT monocyte-derived and resident macrophages have different transcriptome profiles

Using various mouse strains, such as CX₃CR1^+/GFP (Fig. 1A) and CX₃CR1^creER/+ ROSA^tdTomato (Supplementary Fig. 1A and Fig. 1B) mice, and techniques like parabiosis (Fig. 1C) and genetic fate mapping[44], we observed that CX₃CR1⁺ CCR2⁺ and CX₃CR1⁻ CCR2⁻ VAT macrophages are monocyte-derived and tissue-resident, respectively. We performed a lineage tracing experiment in CX₃CR1-CreER-TdTomato mice to probe the conversion of CX₃CR1⁺ CCR2⁺ macrophages into CX₃CR1⁻

CCR2⁻ macrophages (Supplementary Fig. 1B). Four weeks after tamoxifen injection, most tdTomato⁺ macrophages were CX₃CR1^YFP⁺. Only about 12% of them were CX₃CR1^YFP⁻. These data indicate that CX₃CR1⁺ macrophages convert to CX₃CR1⁻ macrophages, albeit at low levels. RNA sequencing revealed that these two subsets of macrophages express discrete transcriptome profiles (Fig. 1D and Supplementary Fig. 1C) (GSE 118226)[44]. CX₃CR1⁺ CCR2⁺ monocyte-derived macrophages expressed genes encoding inflammatory molecules such as matrix metalloproteinases, tumor necrosis factor superfamily 14, and IFNB1 (Supplementary Fig. 1C) and other pro-inflammatory genes typically expressed by monocyte-derived macrophages (Fig. 1E)[72–79]. Further, CX₃CR1⁻ CCR2⁻ VAT resident macrophages expressed genes involved in collagen synthesis (Supplementary Fig. 1C) and other genes typically expressed by tissue-resident macrophage (Fig. 1E)[74,80–85]. Analysis of publicly available bulk RNA sequencing data (GSE112396)[86] revealed that anti-inflammatory CD206⁺ and pro-inflammatory CD206⁻ VAT macrophages exhibited a similar gene signature (Fig. 1F). Consistent with their pro-inflammatory gene expression, CX₃CR1⁺ CCR2⁺ and CD206⁻ macrophages expressed high levels of the genes, which positively correlate with insulin resistance (Fig. 1G, H and Supplementary Fig. 1D)[87–91]. In contrast, CX₃CR1⁻ CCR2⁻ and CD206⁺ macrophages expressed similar transcriptomic profiles including higher levels of insulin sensitivity and anti-inflammatory genes[87,91–96] (Fig. 1G–J and Supplementary Fig. 1D, E). Altogether, these data demonstrate that VAT CX₃CR1⁺ CCR2⁺ macrophages are pro-inflammatory while CX₃CR1⁻ CCR2⁻ macrophages are anti-inflammatory and express high levels of the genes, which promote systemic insulin sensitivity.

### Chronic inflammation leads to apoptosis of VAT resident macrophages

The balance between monocyte-derived infiltrating and tissue-resident macrophages in metabolic tissues is a key contributing factor in the maintenance of metabolic health. We first aimed to assess the changes in the macrophage subsets in VAT in chronic inflammatory diseases such as obesity. We observed an increase in CCR2⁺ monocyte-derived macrophages and a decrease in CCR2⁻ VAT resident macrophages in omental adipose tissue of obese patients (Fig. 1K, L and Supplementary Fig. 1F). The changes in these macrophage subsets followed a similar pattern in obese mouse VAT (Fig. 1M and Supplementary Fig. 1G–I). We also calculated the frequencies and numbers of CX₃CR1⁻ CCR2⁺ macrophages in lean and obese mice (Supplementary Fig. 1J). Although the frequencies of these macrophages decreased, their numbers were not significantly changed in obese mice. Furthermore, we also assessed CD9-expressing lipid-associated macrophages and TIM4⁺ VAT resident macrophages at one and two months after HFD initiation. The numbers of both subsets of macrophages increased significantly at one month after HFD (Supplementary Fig. 1K). However, although CD9-expressing lipid-associated macrophages remained significantly increased at two months after HFD, the abundance of TIM4⁺ VAT resident macrophages diminished at this time. We further investigated the dynamics of VAT resident macrophages at different time points after HFD feeding in CX₃CR1^creER/+ ROSA^tdTomato mice using intravital microscopy (Fig. 1N). The frequency of monocyte-derived macrophages (tdTomato⁺) increased whereas the proportions of adipose resident macrophages (tdTomato⁻) decreased at 2 and 4 months after HFD initiation (Fig. 1O and Supplementary videos 1 and 2).

Further, to understand if apoptosis contributed to the decline in CX₃CR1⁻CCR2⁻ tissue-resident macrophage numbers in obese mice, we measured annexin V, a late apoptosis marker, and caspase 3, an early apoptosis marker, in this macrophage population. Obese mice had augmented proportions of annexin V⁺ (Fig. 1P and Supplementary Fig. 1L) and caspase 3⁺ (Fig. 1Q, R and Supplementary Fig. 1M, N) resident macrophages in VAT compared to lean mice, indicating a higher rate of apoptosis in this macrophage subset in obesity. In contrast, the frequency of caspase 3⁺ CCR2⁺ macrophages did not significantly

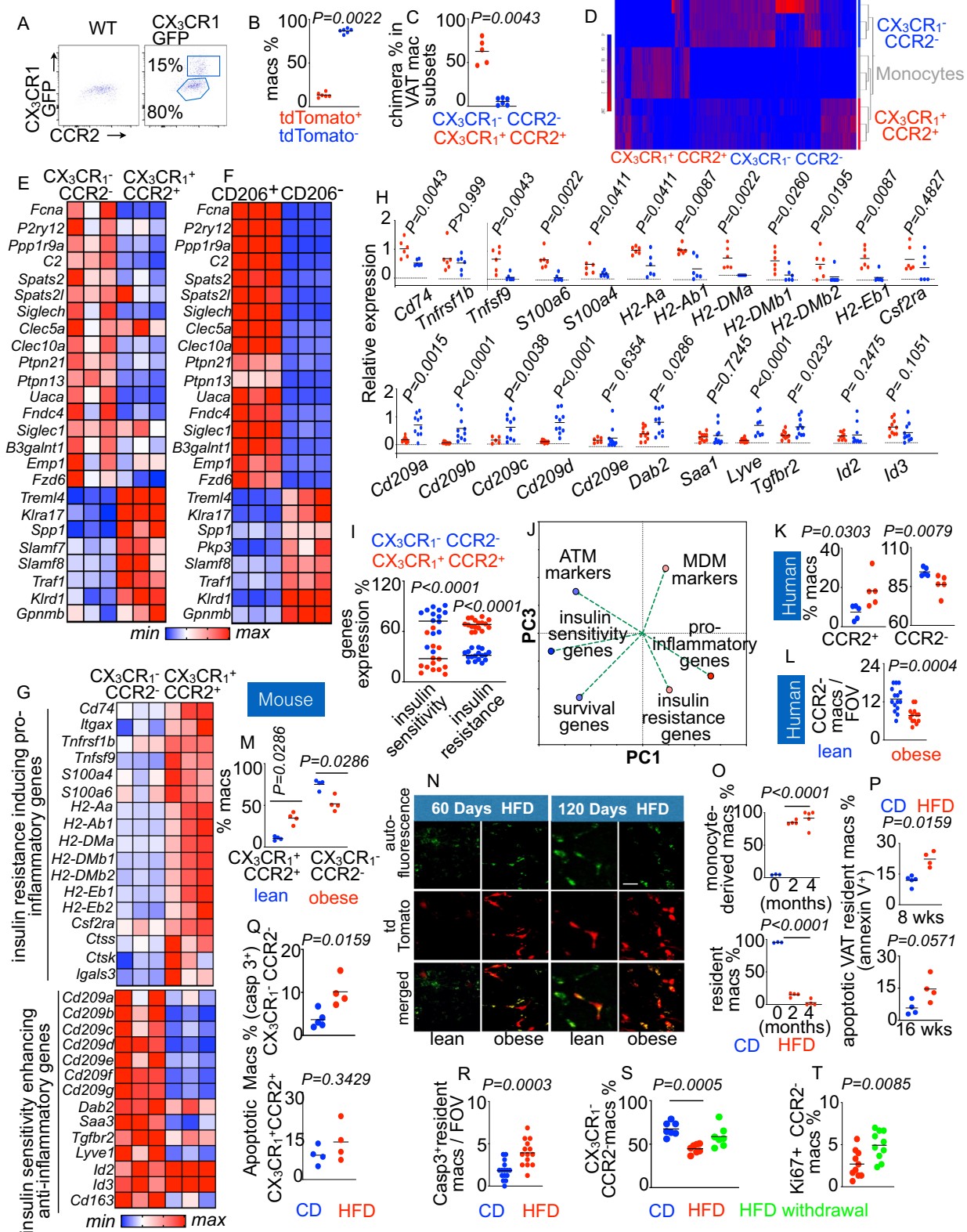

change in obese mice (Fig. 1Q). We next determined whether the change in the frequency of the VAT macrophage subsets after HFD feeding is permanent. As expected, HFD feeding significantly increased monocyte-derived macrophages in VAT while the resident macrophage population was decreased (Supplementary Fig. 1S and O). Interestingly, the frequencies of the macrophage subsets were restored two months after HFD withdrawal (Fig. 1S and Supplementary

Fig. 1O), indicating that the alteration in the numbers of the macrophage subsets with an HFD is reversible. To ascertain whether the emergence of newly generated VAT resident macrophages after HFD withdrawal is due to their local proliferation or higher differentiation of infiltrated monocytes, we carried out the following experiments. We performed parabiosis between two congenically different mice and quantified the frequency of VAT resident macrophages derived from

**Fig. 1 | VAT monocyte-derived and resident macrophages have different transcriptomic profiles. A** Flowcytometric analysis of the macrophage subsets in visceral adipose tissue of CX$_3$CR1$^{+/GFP}$ mice which express GFP under the CX$_3$CR1promoter. **B** tdTomato$^+$ (CX$_3$CR1$^{high}$) macrophages were quantified at day 7 after a single i.p injection of tamoxifen in CX$_3$CR1$^{CreER/+}$ ROSA$^{tdTomato/+}$ mice that express YFP under the CX$_3$CR1 promoter and tdTomato in CX$_3$CR1-expressing cells upon tamoxifen injection using intravital microscopy (n = 6/group). **C** Parabiosis between C57BL/6 and CX$_3$CR1$^{GFP/+}$ mice was performed. Flow cytometry was conducted to enumerate chimerism in the macrophage subsets in the C57BL/6 mice six months after parabiosis. (n = 5 for CX$_3$CR1$^-$ and 6 for CX$_3$CR1$^+$). **D** Heatmap displaying the genes with at least a two-fold difference between the VAT macrophage subsets and with FDR < 0.01 (n = 3/group). **E–G** Heatmaps displaying the expression of the genes using bulk RNA sequencing comparing CX$_3$CR1$^+$CCR2$^+$ and CX$_3$CR1$^-$ CCR2$^-$ macrophages (n = 3/group) and CD206$^-$ and CD206$^+$ macrophage subsets of VAT (n = 3/group). **H** q-PCR quantification of the genes associated with glycemia and diabetes in CX$_3$CR1$^+$CCR2$^+$ and CX$_3$CR1$^-$ CCR2$^-$ macrophages sorted from VAT of lean mice (n = 6–12/group). **I** Bar graph representing the frequency of CX$_3$CR1$^+$CCR2$^+$ and CX$_3$CR1$^-$ CCR2$^-$ VAT macrophages enriched in the insulin sensitivity and resistance genes shown in (**G, H**) (n = 3-6 /group). **J** PCA plot showing the relations among the genes responsible for insulin sensitivity, survival, resident macrophage (ATM) markers, inflammation, insulin resistance, and monocyte-derived macrophages (MDM) markers in the VAT macrophage subsets using bulk RNA sequencing. **K, L** Frequencies of CCR2$^+$ and CCR2$^-$ macrophage subsets in human VAT as measured by flow cytometry (**K**) (n = 5/group) and confocal microscopy (**L**) (n = 15 for lean and 13 for obese). **M–O** Quantification of the VAT macrophage subsets in HFD-fed mice by flow cytometry (n = 4/group) (**M**) and serial intravital microscopy (Scale bar = 10 μm) (**N, O**) was performed in lean and obese CX$_3$CR1$^{CreER/+}$ ROSA$^{tdTomato/+}$ mice (n = 3 for CD, 4 for HFD 2 months, and 5 for HFD 4 months). **P–R** Apoptosis in VAT resident and monocyte-derived macrophages in lean and obese mice was examined using annexin V by flow cytometry (**P**) (n = 4/group), and caspase 3 staining by flow cytometry (**Q**) (n = 5 for CD and 4 for HFD/group) and confocal microscopy (**R**) (n = 14/group). **S** Quantification of the VAT macrophage subsets in lean and obese CX$_3$CR1$^{creER/+}$ ROSA$^{tdTomato}$ mice before and after removal of HFD (n = 7/group). **T** Evaluation of Ki-67$^+$ VAT resident macrophages after HFD withdrawal (n = 10/group). Mean ± s.e.m. *P < 0.05, ** P < 0.01, ***P < 0.001. The Mann–Whitney test (two-tailed) was used to determine the significance between two groups. One-way ANOVA with Bonferoni's post hoc correction test was performed to determine differences among data obtained from more than two groups (Fig. 1O and S).

the parabiont partner three months after HFD initiation, and two and four months after withdrawal of the diet (Supplementary Fig. 1P). Parabiosis allows us to discern if a cell population is derived from circulating hematopoietic cells. HFD withdrawal did not significantly change parabiont-derived VAT resident macrophage frequencies, ruling out the possibility of differentiation of recruited monocytes into this macrophage subset after the diet withdrawal. VAT resident macrophages after HFD withdrawal exhibited augmented Ki-67 staining (Fig. 1T and Supplementary Fig. 1Q), indicating their higher proliferation after HFD withdrawal.

## Loss of VAT resident macrophages enhances insulin resistance in diet-induced obesity

As CX$_3$CR1$^-$CCR2$^-$ tissue-resident macrophages are enriched for insulin sensitivity genes, we were interested in deciphering the functions of these macrophages in diet-induced insulin resistance. To this end, we generated LysM$^{cre/+}$ Chr2$^{fl/fl}$ mice, which express the channelrhodopsin-2 (ChR2) protein in macrophages after exposure to blue light (450–490 nm) (Fig. 2A). Photostimulation of this protein leads to prolonged action potential firing activity, resulting in cellular apoptosis[44]. Extended blue light exposure (BLE) resulted in the reduction of VAT-resident macrophages but not monocyte-derived macrophages possibly due to a high turnover and increased apoptosis of the latter cell population compared to the white light exposure (WLE) (Fig. 2B). Indeed, blue light exposure in obese LysM$^{cre/+}$ ChR2$^{fl/fl}$ mice significantly enhanced Annexin V expression, a marker of early apoptosis, in CX$_3$CR1$^+$ CCR2$^+$ VAT macrophages (Supplementary Fig. 2A). We next examine if blue light exposure also affects TIM4$^+$ adipose resident macrophages and CD9$^+$ lipid-associated macrophages (LAM), which are crucial for adipose biology[43,97]. We observed that TIM4$^+$ macrophage abundance was lower in VAT of LysM$^{cre/+}$ ChR2$^{fl/fl}$ mice after blue light exposure as expected (Supplementary Fig. 2B). Moreover, the number of CD9$^+$ LAMs, which are recruited in obesity and play protective roles[43,97], diminished significantly upon blue light exposure. Additionally, we have examined if blue light exposure in VAT of the optogenetic mice alters the expression of Annexin V, an early marker of apoptosis, in macrophages residing in distant organs. This treatment did not significantly change Annexin V expression in microglia, Kupffer cells, and splenic macrophages (Supplementary Fig. 2C). Consistently, we observed no significant difference in total hepatic macrophage and Kupffer cell numbers (Supplementary Fig. 2D). VAT-resident macrophage depletion resulted in glucose intolerance and insulin resistance (Fig. 2C and Supplementary Fig. 2E). Consistently, these mice had higher levels of triglycerides, free fatty acids, and free glycerol (Fig. 2C). Furthermore, pAkt and glut4 expression, which is associated with insulin sensitivity, was impaired after blue light exposure (Fig. 2D and Supplementary Fig. 2F, G). The levels of Slc2a4, Irs1, 2 and 3, Sfrp5, Adpn, and Ces1e, which promote glucose clearance[98–104], were significantly decreased after VAT resident macrophage depletion with blue light illumination (Fig. 2E). Transcript analysis further showed that blue light-exposed VAT samples exhibited higher levels of proinflammatory cytokines associated with obesity-mediated insulin resistance (Fig. 2F)[105,106]. Collectively, these data suggest that VAT resident macrophages improve insulin sensitivity.

## CX$_3$CR1$^-$ CCR2$^-$ VAT resident macrophages express SerpinB2 at high levels

We noticed that CX$_3$CR1$^-$ CCR2$^-$ macrophages expressed high levels of Pdgfc, Gata1, and Gata2 that promote tissue-resident macrophage survival by inhibiting the activation of caspases and subsequent apoptosis[107–110] (Fig. 3A). Among all cell survival genes, SerpinB2 was highly expressed in VAT resident macrophages (Fig. 3A and Supplementary Fig. 3A). Various organs, such as the liver and muscle, important in glucose homeostasis did not express detectable levels of SerpinB2 (Fig. 3B). Among all cells in adipose tissue, which we tested, macrophages expressed very high levels of SerpinB2 (Fig. 3C). Macrophages differentiated from THP-1 human monocytes using phorbol-12-myristate-13-acetate also exhibited high SerpinB2 expression; however, SerpinB2 was not detectable in the cell culture conditioned medium (Fig. 3D) although copious amounts of pro-inflammatory cytokines were present in the culture, suggesting that SerpinB2 is poorly secreted[49]. To validate the RNA sequencing data, we measured SerpinB2 expression in the macrophage subsets in gonadal adipose tissue of lean mice. Adipose tissue-resident macrophages expressed significantly higher levels of SerpinB2 mRNA and protein (Fig. 3E–G) than monocyte-derived macrophages. To discern if SerpinB2 is also expressed at high levels in human adipose tissue-resident macrophages, we collected human omental adipose tissue. Human CCR2$^-$ macrophages also expressed higher levels of SerpinB2 mRNA and protein than CCR2$^+$ macrophages (Fig. 3H–J).

## IFN-γ-mediated SerpinB2 downregulation leads to VAT resident macrophage apoptosis in chronic inflammation

We next sought to delineate the mechanisms behind enhanced apoptosis of VAT resident macrophages in chronic inflammation precipitated by obesity. To determine if SerpinB2 levels in VAT resident macrophages alter in obesity, we first quantified SerpinB2 in

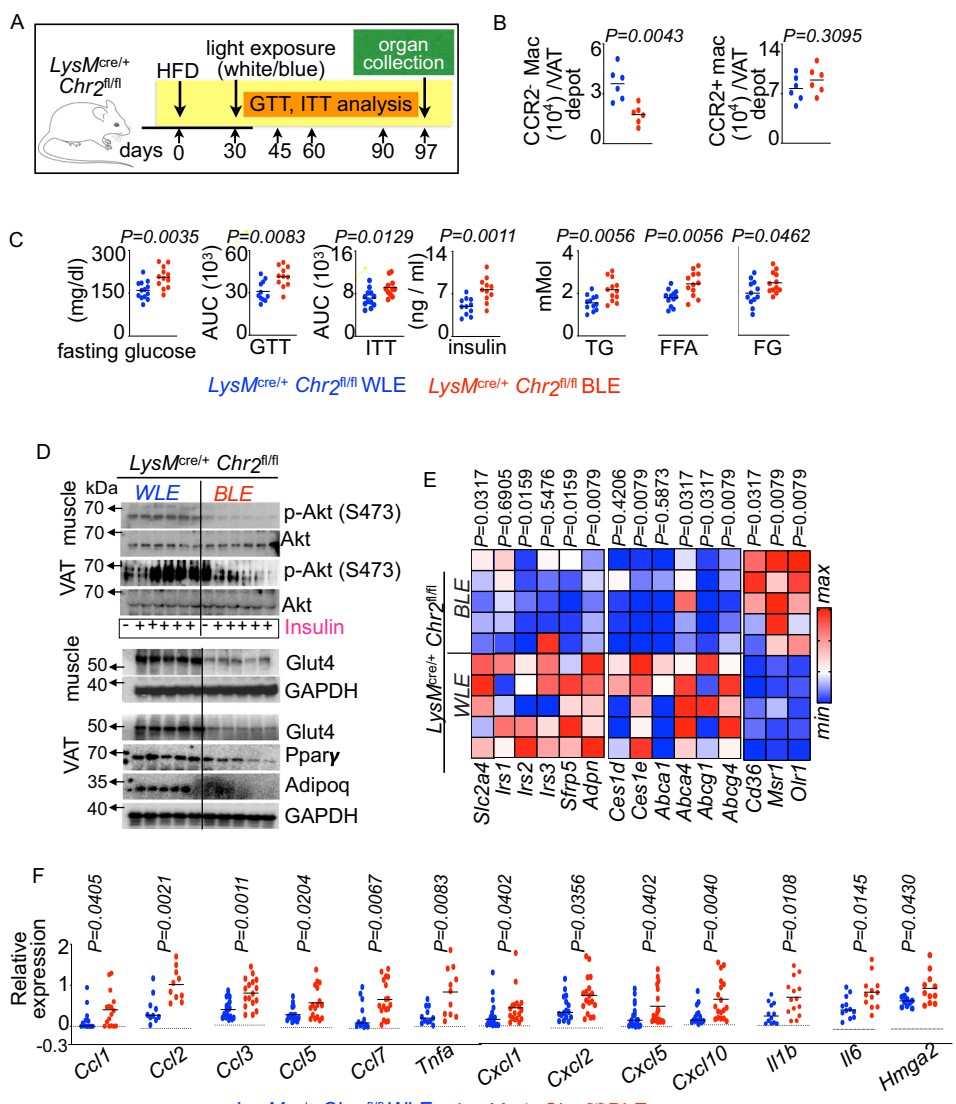

**Fig. 2 | VAT resident macrophage loss exacerbates obesity-induced metabolic complications. A–F** Blue/white light-exposed (BLE and WLE, respectively) *LysM*cre/+ *Chr2*fl/fl mice were fed with an HFD. **A** Schematic diagram showing the experimental design. **B** Enumeration of the VAT macrophage subsets using flow cytometry (n = 6/group). **C** GTT and ITT were performed. The concentrations of fasting blood glucose, serum insulin, triglycerides (TG), free fatty acids (FFA), and free glycerol (FG) were evaluated (n = 13 for WLE and 12 for BLE, combined data of 2 independent experiments). **D** Immunoblot images of the represented proteins in muscle and VAT (n = 5–6/group). **E, F** qPCR quantification of the expression of the indicated metabolic and inflammatory genes measured in VAT represented by heat maps (**E**) (n = 5/group) and bar graphs (**F**) (n = 15/group). Mean ± s.e.m. * *P* < 0.05, ** *P* < 0.01. The Mann–Whitney test (two-tailed) was used to determine the significance between two groups.

CCR2⁻ macrophages in omental adipose tissue of healthy and diabetic/obese patients and observed a significant downregulation of SerpinB2 in these macrophages of diabetic/obese patients (Fig. 3K–M and Supplementary Table 1). Furthermore, using bivariate analysis, we observed a significant negative correlation between the frequency of SerpinB2-expressing macrophages in VAT and the BMI of the patients (Fig. 3N). Congruent with the reduction in SerpinB2 expression in VAT CCR2⁻ macrophages in obese humans, gonadal adipose tissue of obese mice harbored fewer SerpinB2-expressing macrophages (Fig. 3O, P). Additionally, adipose tissue-resident macrophages in obese patients and mice downregulated the expression of *SerpinB2* (Fig. 3Q, R). It has been reported that obesity can increase endogenous palmitic acid levels[111]. This excess palmitic acid can promote inflammation[112]. Because exaggerated inflammation can reduce SerpinB2 expression, we sought to test whether palmitic acid can reduce the expression of this gene. Indeed, palmitate treatment in bone

marrow-derived macrophages of mice suppressed SerpinB2 expression (Fig. 3S).

Adipose tissue secretes high levels of pro-inflammatory cytokines such as IFN-γ and TNF in obesity[113–115]. THP1 macrophages treated with IFN-γ, but not TNF, expressed lower levels of SerpinB2 whereas inhibition of IFN-γ with Bay-117082 increased SerpinB2 expression (Fig. 4A, B, and Supplementary Fig. 3B). Consistently, fibroblasts and macrophages produced high levels of IFN-γ in obese mice (Fig. 4C). We hypothesized that IFN-γ binds to its receptor on VAT macrophages to decrease SerpinB2 expression. Consistently, obese mice with myeloid-specific deficiency in the IFN-γ receptor (Ifngr1) had elevated SerpinB2 expression in VAT resident macrophages compared to wildtype control mice (Fig. 4D and Supplementary Fig. 3C). In line with this finding, this macrophage subset in myeloid-specific *Ifngr1*-deficient mice was more abundant and less apoptotic (Fig. 4D, E and Supplementary Fig. 3C–E) while the total number of adipose macrophages did not change in these mice

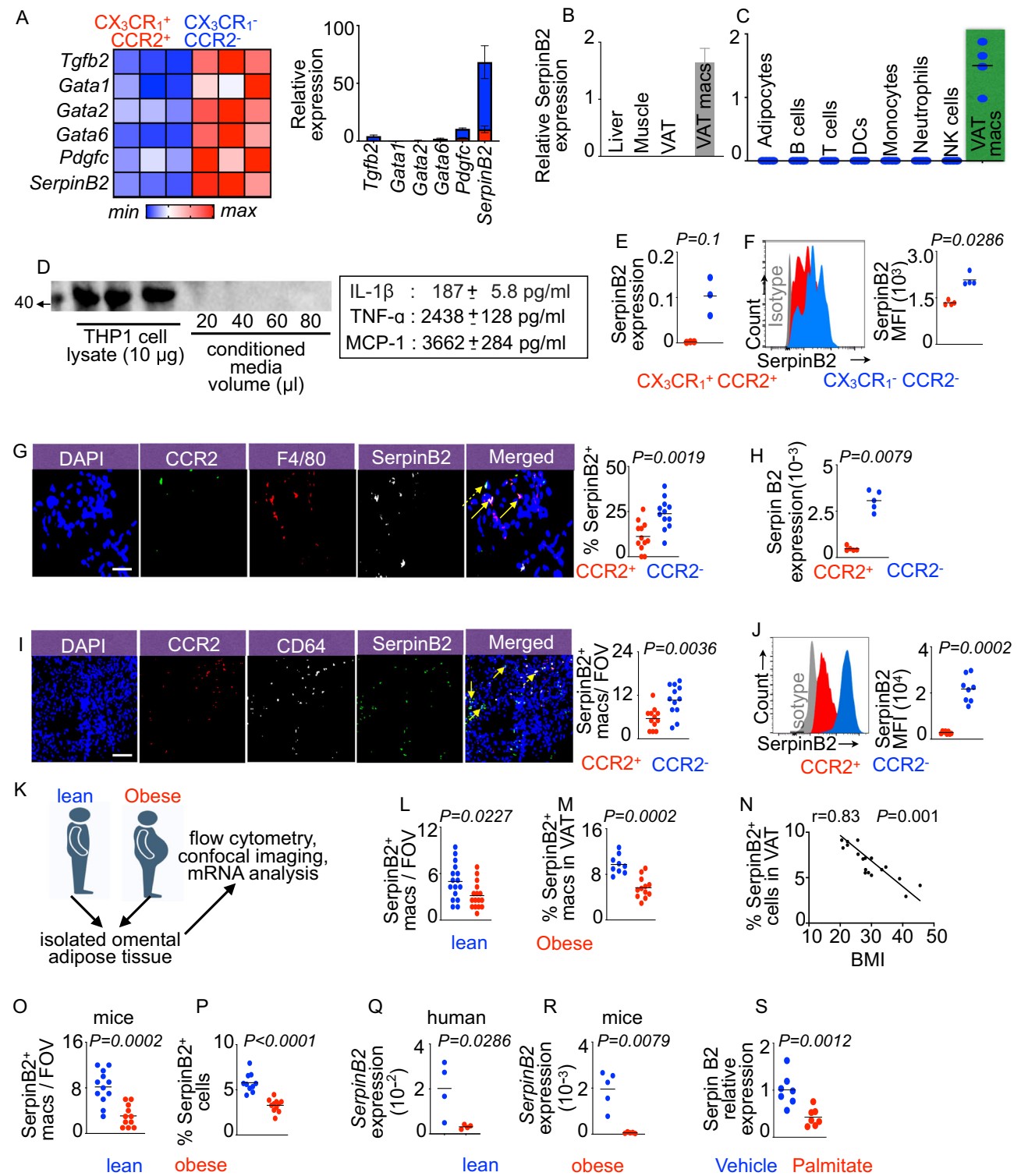

compared to the control mice (Supplementary Fig. 4A). Additionally, *LysM*<sup>cre/+</sup> *Ifngr1*<sup>fl/fl</sup> mice had reduced fasting blood glucose levels, improved glucose tolerance, and reduced levels of triglycerides, free glycerol, and free fatty acids in the serum; however, their body weights were unchanged compared to *LysM*<sup>+/+</sup> *Ifngr1*<sup>fl/fl</sup> mice (Fig. 4F and Supplementary Fig. 3F). *LysM*<sup>cre/+</sup> *Ifngr1*<sup>fl/fl</sup> mice also had higher pAkt, glut4, adiponectin, and Pparγ expression compared to the control group, indicating improved insulin signaling (Fig. 4G, H and Supplementary Fig. 3G). In line with these findings, the expression of insulin sensitivity-promoting genes like *Slc2a4, Irs1, Irs3, Sfrp5, Adipoq, Ces1d, Abca4, Abcg1,*

and *Abcg4* were increased in VAT of these mice (Fig. 4I). In addition, *Cd36, Msr1, Olr1, Ccl3, Ccl5, Cxcl1, Cxcl2, Il1b, Il6,* and *Tnf* transcripts, which are associated with increased risk of obesity and insulin resistance, were markedly diminished in VAT of *LysM*<sup>cre/+</sup> *Ifngr1*<sup>fl/fl</sup> mice (Fig. 4I, J). Our in-silico analysis showed that the *SerpinB2* promoter has binding motifs for Ikaros, a transcriptional suppressor (Fig. 4K and Supplementary Fig. 3H). A chromatin immunoprecipitation assay confirmed the binding of Ikaros to the *SerpinB2* promoter (Fig. 4L), and IFN-γ treatment significantly increased Ikaros expression in BMDM (Fig. 4M). Treatment with lenalidomide, a known inhibitor of Ikaros[116], significantly blocked

**Fig. 3 | CX₃CR1⁻ CCR2⁻ VAT resident macrophages express high levels of SerpinB2. A** The heatmap displays the levels of the cell survival regulating genes in CX3CR1⁻ CCR2⁻ VAT resident macrophages compared to CX3CR1⁺ CCR2⁺ VAT monocyte-derived macrophages using RNA sequencing (n = 3/group).
**B, C** *SerpinB2* expression in various organs (**B**) (n = 4/group) and adipose cells (**C**) (n = 4/group) is assessed using qPCR. **A–C** n represent the number of mice the cells or organs were obtained from. **D** In PMA-differentiated THP-1 macrophages, SerpinB2 levels were measured in the cell lysates and conditioned medium by immunoblot. The proinflammatory cytokines in the conditioned media were quantified by ELISA (n = 12–16/group). **E–G** SerpinB2 quantification in mouse CX₃CR1⁺ CCR2⁺ and CX₃CR1⁻ CCR2⁻ VAT macrophages by qPCR (**E**) (n = 3/group), flow cytometry (**F**) (n = 4/group), and confocal microscopy (**G**) (n = 12/group, each dot represents one tissue section.) in lean mice. The dashed and solid arrows in (**G**) indicate CCR2⁺ and CCR2⁻ macrophages, respectively. Scale bar = 20 µm. **H–J** Serpin B2 expression in human CCR2⁺ and CCR2⁻ VAT macrophages were measured by qPCR (**H**) (n = 5/group, each dot represents one mouse.), confocal imaging (**I**) (n = 12/group), Scale bar = 30 µm, and flow cytometry (**J**) (n = 8/group, each dot represents one mouse). **K** Schematic diagram depicting the experiments performed with patient omental VAT. Created in BioRender. Dutta, P. (2025) https://BioRender.com/0h4nb79. **L, M** SerpinB2-expressing macrophages in lean and obese human VAT were quantified by confocal microscopy (**L**) (n = 16/group) and flow cytometry (**M**) (n = 9 for lean and 12 for obese). **N** Correlation between BMI and SerpinB2⁺ cells in human VAT using confocal imaging (n = 21). **O, P** The frequency of SerpinB2⁺ macrophages in VAT of lean and obese mice by confocal microscopy (**O**) (n = 12 for lean and 11 for obese) and flow cytometry (**P**) (n = 10/group) is discerned. **Q, R** SerpinB2 mRNA was measured in VAT resident macrophages of lean and obese humans (**Q**) (n = 4/group, each dot represents one human sample) and mice (**R**) (n = 5/group) by qPCR. **S** SerpinB2 was quantified by qPCR in BMDM after palmitate treatment (n = 7/group, each dot represents cells cultured in one well.). Mean ± s.e.m. * $P < 0.05$, ** $P < 0.01$, *** $P < 0.001$. The Mann Whitney test (two-tailed) was used to determine the significance between two groups. Linear regression analysis was performed for the data presented in (**D**).

IFN-γ-mediated *SerpinB2* downregulation (Fig. 4N). Congruently, monocyte-derived macrophages, which express SerpinB2 at lower levels, were enriched for the genes encoding the receptors for IFN-γ (Fig. 4O). Altogether, these data indicate that IFN-γ-induced Ikaros decreases *SerpinB2* expression in VAT resident macrophages. To determine whether SerpinB2 is indeed involved in the survival of macrophages, we analyzed apoptosis of BMDM, THP-1 macrophages, and VAT resident macrophages. We observed elevated caspase 3 levels in BMDM lacking *SerpinB2* (Fig. 4P) and decreased proportions of apoptotic THP-1 macrophages after SerpinB2 overexpression (Fig. 4Q). Furthermore, obese *SerpinB2⁻/⁻* mice harbored an augmented frequency of apoptotic VAT resident macrophages (Fig. 4R). In contrast, the numbers of Kupffer cells in obese *SerpinB2⁺/⁺* and *SerpinB2⁻/⁻* mice were similar (Supplementary Fig. 4B). In aggregate, these data suggest that SerpinB2-deficient macrophages are apoptotic.

### Increased mitochondrial ROS in *SerpinB2⁻/⁻* macrophages induces apoptosis

Next, we investigated the mechanisms of enhanced apoptosis in *SerpinB2⁻/⁻* macrophages. To understand how SerpinB2 mediates macrophage survival, we measured cytochrome c levels in cytosol and mitochondria. The release of cytochrome c from the mitochondria into the cytosol has been reported to be a major mechanism of cellular apoptosis[117–119]. We observed increased concentrations of cytosolic cytochrome c while the levels of mitochondrial cytochrome c decreased in *SerpinB2⁻/⁻* BMDM compared to *SerpinB2⁺/⁺* BMDM (Fig. 5A, B). These findings suggest that loss of SerpinB2 results in cytochrome c release from the mitochondria into the cytosol. Cytochrome c release into the cytosol depends on mitochondrial oxidative stress[118]. To understand the mechanisms of cytochrome c translocation from mitochondria to the cytoplasm in the absence of SerpinB2, we quantified the genes encoding antioxidant enzymes in the VAT macrophage subsets. We ascertained that CX₃CR1⁻ CCR2⁻ VAT macrophages of lean mice were enriched with anti-oxidant enzymes such as superoxide dismutase (*Sod3*), glutathione-S-transferases (*Gstm, Gsta, Gstp*, and *Gstt*), mitochondrial membrane-associated *Mgst1*, metallothioneins (*Mt1* and *Mt2*), and mitochondrial specific anti-oxidant enzymes like glutathione peroxidase (*Gpx7*), malic enzyme 1 (*Me1*), and isocitrate dehydrogenase2 (*Idh2*) (Fig. 5C, D)[120]. Similarly, CX₃CR1⁻ CCR2⁻ VAT macrophages had elevated expression of the GSH encoding genes (Fig. 5E). Additionally, the expression of the GSH-encoding genes in CX₃CR1⁻ CCR2⁻ VAT macrophages was decreased in obesity (Fig. 5F). Moreover, GSH gene expression in CX₃CR1⁻ CCR2⁻ VAT macrophages and glucose clearance were correlated (Fig. 5G). Taken together, these results demonstrate that CX₃CR1⁻ CCR2⁻ VAT macrophages are enriched in GSH in lean mice. Among the genes involved in GSH precursor import, only *Slc7A13*, which mediates the transport of glutamic acid, one of the precursors for GSH synthesis,

was upregulated in CX₃CR1⁻ CCR2⁻ VAT macrophages (Fig. 5H). Notably, SerpinB2 deficiency significantly curtailed the levels of these antioxidant genes in VAT macrophages (Fig. 5I). Consistently, *SerpinB2⁻/⁻* BMDM isolated from lean or obese mice had higher levels of mitochondrial ROS than *SerpinB2⁺/⁺* BMDM (Fig. 5J, K and Supplementary Fig. 4C, D). Mitotracker green MFI was significantly decreased in *SerpinB2⁻/⁻* BMDM treated with palmitate, indicating lower mitochondrial volume in these cells compared to *SerpinB2⁺/⁺* BMDM (Supplementary Fig. 4E). To examine the function of mitochondrial ROS in cytochrome c release into the cytosol from the mitochondria, we treated BMDM derived from *SerpinB2⁺/⁺* and *SerpinB2⁻/⁻* mice with mitotempol, a mitochondrial ROS scavenger. This treatment significantly decreased cytosolic cytochrome c levels while the concentrations of mitochondrial cytochrome c were increased in *SerpinB2⁻/⁻* BMDM (Fig. 5L) indicating suppressed cytochrome c release from the mitochondria. Concomitantly, the treatment caused a non-significant decrease in the level of annexin V, an early apoptosis marker, in *SerpinB2⁻/⁻* BMDM (Fig. 5M). Nevertheless, annexin V mean fluorescent intensities in *SerpinB2⁺/⁺* BMDM were unchanged after the treatment (Fig. 5M). To understand if SerpinB2 deficiency alters the inflammasome, which can trigger pyroptotic cell death, we measured *Nlrp3*, which is involved in inflammasome signaling. The expression of *Nlrp3* in *SerpinB2⁺/⁺* and *SerpinB2⁻/⁻* adipose resident macrophages was similar (Supplementary Fig. 4F), suggesting that this pathway does not contribute to increased apoptosis of this macrophage subset in absence of *SerpinB2*. Altogether, these data imply that the lack of SerpinB2 in macrophages elevated mitochondrial ROS, which triggered cytochrome c release into the cytosol.

### *SerpinB2* deficiency aggravates inflammation

Adipose tissue inflammation mediated by macrophages contributes to insulin resistance[39,45]. OXPHOS analysis using SeaHorse XF bioanalyzer in BMDM obtained from lean and obese C57BL/6 mice showed that BMDM of obese mice have elevated oxygen consumption rates (Fig. 6A and Supplementary Fig. 5A). Interestingly, OCR and inflammatory cytokines are strongly correlated (Fig. 6A). From these observations, we hypothesized that enhanced OXPHOS activity in macrophages exaggerates inflammation in obesity. Consistently, obese *LysMᶜʳᵉ/⁺ COX10ᶠˡ/ᶠˡ* mice that lack the mitochondrial complex IV activity[121–125] exhibited decreased inflammation, and improved glucose clearance, insulin sensitivity, and insulin levels compared to obese *LysM⁺/⁺ COX10ᶠˡ/ᶠˡ* mice although the body weights were similar between the groups (Fig. 6B and Supplementary Fig. 5B, C). We observed depressed basal and maximal respiration and spare capacity while extracellular acidification rates were unchanged in *Cox10*-deficient BMDM compared to WT BMDM (Supplementary Fig. 5D, E). Similarly, SerpinB2 deficiency increased OCR (Supplementary Figs. 6C and 5F) and inflammation (Supplementary Figs. 5G and H) in BMDM while SerpinB2 overexpression in THP-1

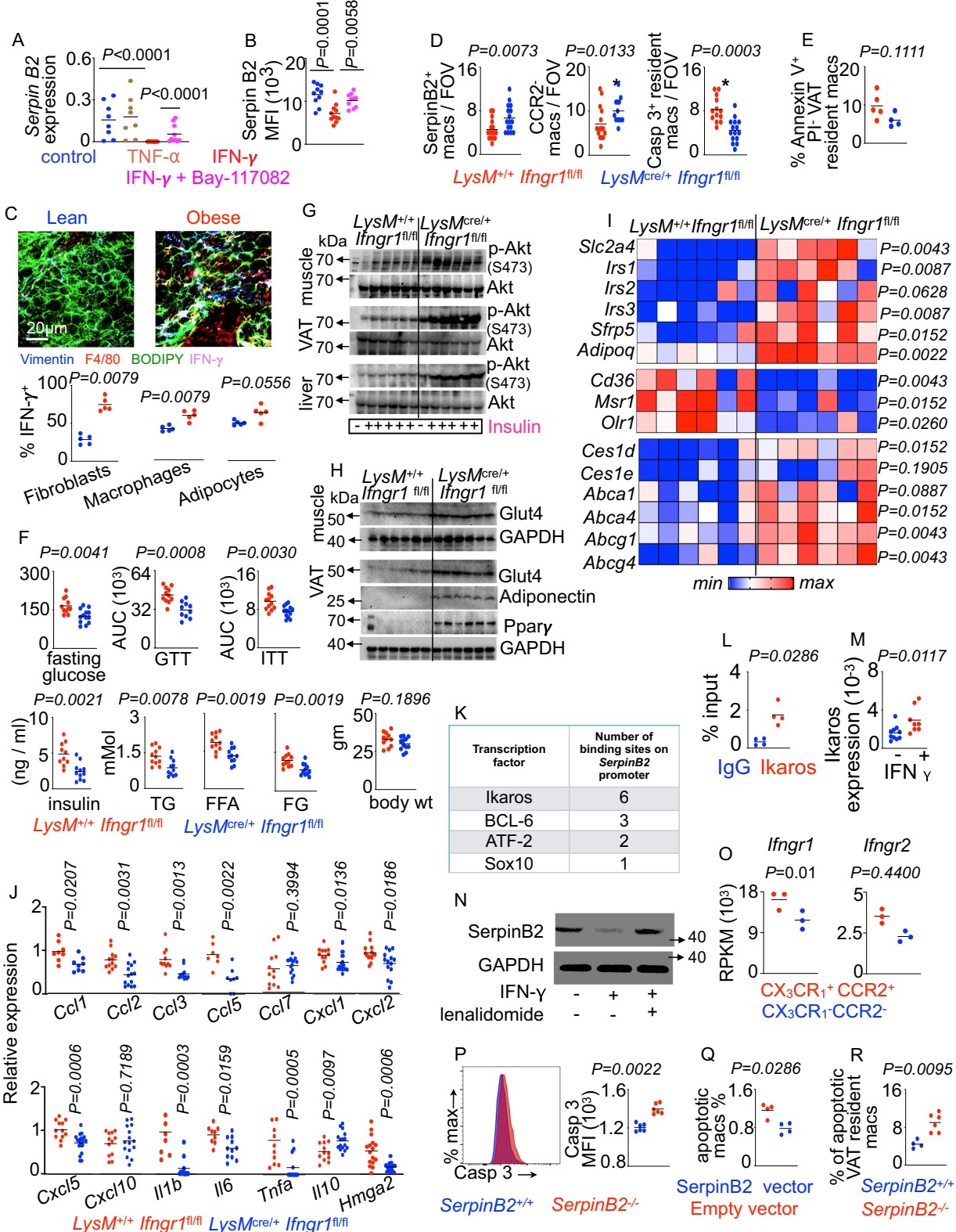

macrophages depressed inflammation (Fig. 6D). In line with these findings, we observed that *SerpinB2*−/− adipose macrophages expressed higher levels of inflammatory genes, such as *Il1b, Tnf, Ccl2, Ccl3, iNOS*, etc., and lower levels of pro-resolving genes, such as *Il10* and *Arg1*, than *SerpinB2*+/+ adipose macrophages (Supplementary Fig. 5I, J). In aggregate, these data support the premise that SerpinB2 is essential for suppressing macrophage-mediated inflammation.

## SerpinB2 deficiency in macrophages exacerbates glucose intolerance in diet-induced obesity

Our data demonstrate that VAT resident macrophages are crucial to the maintenance of insulin sensitivity (Fig. 1). Because SerpinB2 expression is critical for the survival of this macrophage subset, and there is an inverse correlation between macrophage-specific SerpinB2 expression in omental adipose tissue and BMI in patients, we

**Fig. 4 | IFN-γ-mediated SerpinB2 downregulation leads to resident VAT macrophage apoptosis in obesity. A, B** SerpinB2 was quantified in THP-1 macrophages after stimulation with IFN-γ in the presence or absence of Bay-118072, an NF-kB inhibitor, by qPCR (**A**) (n = 8/group) and confocal imaging (**B**) (n = 10/group). **A, B** each dot represents cells cultured in one well. **C** VAT from lean and obese mice were stained for the markers of fibroblasts, macrophages, and adipocytes along with IFN-γ. The frequencies of IFN-γ-expressing cells were assessed by confocal microscopy (n = 5/group) Scale bar=20 μm. **D–J** *LysM*^+/+ *Ifngr1*^fl/fl and *LysM*^cre/+ *Ifngr1*^fl/fl mice were fed with an HFD for four months. **D** VAT resident macrophages, SerpinB2-expressing macrophages, and caspase 3⁺ resident macrophages were measured by confocal microscopy (n = 12/group). **E** The frequencies of apoptotic (annexin V⁺ PI⁻) macrophages were measured by flow cytometry (n = 5 for WT and 4 for KO). **F** GTT and ITT were performed, and fasting blood glucose, serum insulin, lipid concentrations, and body weights were evaluated (n = 10/group, combined data of at least 2 independent experiments). **G, H** Immunoblot showing pAkt, total Akt, Glut4, adiponectin, and Ppar*y* expressions in muscle, visceral adipose. and liver (n = 6/group). **I, J** qPCR was

carried out to measure the expression of the metabolic and inflammatory genes in VAT (n = 7/group). **K, L** The top four transcription factors predicted to bind to the SerpinB2 promoter (**K**), and ChIP validation of Ikaros binding to the SerpinB2 promoter (**L**) (n = 4/group, with two sets of primers) are shown. **M** Ikaros quantification by qPCR in BMDM cultured in the presence or absence of 250 milli units IFN-γ (n = 10 and 8 for with and without IFN-γ, respectively). **L, M** each dot represents cells cultured in one well. **N** SerpinB2 expression in BMDM treated with or without IFN-γ and lenalidomide (representative image of the 3 independent experiments). **O** *Ifngr1* and *Ifngr2* were quantified by RNA seq in the adipose macrophage subsets (n = 3/group, each dot represents one mouse.). **P–R** Evaluation of apoptosis in BMDM lacking SerpinB2 (**P**) (n = 6/group), THP-1 macrophages overexpressing *SerpinB2* (**Q**) (n = 4/group), and VAT resident macrophages in obese *SerpinB2*^+/+ and *SerpinB2*^-/- mice (**R**) (n = 4 for WT and 6 for KO) by flow cytometry. * *P* < 0.05, ** *P* < 0.01, ***P* < 0.001. The Mann Whitney test (two-tailed) was used to determine the significance between two groups. One-way ANOVA with Bonferoni's post hoc correction test was performed to determine differences among data obtained from more than two groups (Fig. 4B).

hypothesize that the loss of SerpinB2 expression in macrophages results in glucose intolerance. A glucose tolerance test revealed that *SerpinB2*^-/- mice had delayed glucose clearance and higher fasting insulin levels compared to *SerpinB2*^+/+ mice whereas the body weights were unaltered (Fig. 6E and Supplementary Fig. 6A). To specifically test the contributions of myeloid-specific SerpinB2, we generated the mice that lack SerpinB2 in myeloid cells (Supplementary Fig. 6B). Glucose and insulin tolerance tests revealed that these mice had delayed glucose clearance and aggravated insulin resistance, respectively, compared to age-matched control mice while the body weights of the mice in both groups were similar (Fig. 6F and Supplementary Fig. 6C, D). *SerpinB2* deficiency in myeloid cells also led to increased levels of fasting serum insulin, triglycerides, free fatty acids, and free glycerol suggesting increased whole-body insulin resistance (Fig. 6F). These findings were further confirmed by a hyperinsulinemic-euglycemic clamp study. During this experiment, the glucose infusion rate to match plasma glucose levels was significantly less in *LysM*^cre/+ *SerpinB2*^fl/fl mice (Fig. 6G), demonstrating whole body insulin resistance in these mice. Additionally, plasma insulin and free fatty acid levels were higher in *LysM*^cre/+ *SerpinB2*^fl/fl mice (Fig. 6G) without noticeable differences in the body weights, steady-state whole-body glucose uptake measurements, and fasting rates of endogenous (hepatic) glucose production (Supplementary Fig. 6E). However, hepatic glucose production during steady-state was markedly greater in *LysM*^cre/+ *SerpinB2*^fl/fl mice during the clamp, indicating that the failure to suppress hepatic glucose production accounted for the differences in whole-body insulin sensitivity between the groups (Fig. 6G). In line with this, we observed decreased p-Akt in VAT and muscle (Fig. 6H and Supplementary Fig. 6F), diminished Glut4, Pparγ, and adiponectin in VAT, and reduced Glut4 in muscle (Fig. 6I and Supplementary Fig. 6F). Furthermore, insulin sensitizing *Slc2a4, Irs1, Irs2, SfrpS, Adipoq*, and *Abcg4* transcripts were repressed in VAT of *LysM*^cre/+ *SerpinB2*^fl/fl mice (Fig. 6J). Furthermore, the lack of SerpinB2 in myeloid cells elevated pro-inflammatory gene expression in VAT (Fig. 6J). Similarly, sorted macrophages from VAT from these mice had suppressed expression of anti-inflammatory genes and heightened pro-inflammatory genes (Fig. 6K). SerpinB2 deficiency augmented apoptosis of VAT resident but not monocyte-derived macrophages (Supplementary Fig. 6G). Altogether, these data corroborate the significance of macrophage *SerpinB2* expression in maintaining systemic insulin sensitivity.

## IL-4 reverses exaggerated diet-induced insulin resistance in the absence of myeloid SerpinB2

To investigate the molecular mechanisms of insulin sensitivity exerted by VAT resident macrophages, we assessed the expression of the cytokines promoting insulin sensitivity from the RNA sequencing

analysis data of sorted CX₃CR1⁺ CCR2⁺ and CX₃CR1⁻ CCR2⁻ VAT macrophages. We observed that VAT-resident macrophages expressed higher levels of *Il4, Il33, and Il34* (Fig. 6L). To discern whether increased insulin resistance in *LysM*^cre/+ *SerpinB2*^fl/fl mice is due to lack of IL-4, we supplemented these mice with IL-4 (Supplementary Fig. 6C). IL-4 supplementation accelerated glucose clearance, encouraged insulin sensitivity, and decreased fasting glucose concentrations (Fig. 6F and Supplementary Fig. 6D) while no noticeable improvements in insulin, triglycerides, and free glycerol levels were observed (Fig. 6F). Consistent with these findings, IL-4 replenishment boosted p-Akt, Glut4, and Pparγ expression in VAT, and pAkt and glut4 levels in muscle (Fig. 6H, I and Supplementary Fig. 6H). As expected, the expression of insulin sensitivity-promoting genes was enhanced while the transcripts of the proteins stimulating inflammation and insulin resistance were drastically reduced in VAT (Fig. 6J). Altogether, these findings reveal that IL-4 can reverse aggravated insulin resistance in the absence of myeloid SerpinB2.

## Loss of adipose resident macrophages provokes VAT expansion in diet-induced obesity

Because VAT expansion in obesity is associated with insulin resistance, we deciphered the role of adipose resident macrophages in VAT growth. Towards this end, we depleted VAT resident macrophages in *LysM*^cre/+/*Chr₂*^fl/fl mice with blue light exposure. Blue light-exposed *LysM*^cre/+/*Chr₂*^fl/fl mice had heavier VAT, bigger adipocytes, and higher expression of adipose expansion genes, reflecting the adipocyte enlargement in the absence of VAT resident macrophages (Fig. 7A). In agreement with these data, myeloid-specific SerpinB2 deficiency, which we found to deplete the adipose resident macrophage pool, heightened VAT weight and adipocyte diameter whereas IL-4 replenishment in *LysM*^cre/+ *SerpinB2*^fl/fl mice restored these parameters (Fig. 7B). Consistent with the finding that *Ifngr1* deficiency prevents adipose resident macrophage apoptosis in obesity and improves insulin sensitivity, mice lacking this receptor in myeloid cells had lower VAT weights and smaller adipocytes (Supplementary Fig. 7A). Interestingly, adipose resident macrophages expressed low levels of *Il1b*, which stimulates adipocyte growth[126], and high levels of *Tgfb2*, which inhibits adipocyte growth[127], compared to monocyte-derived macrophages (Supplementary Fig. 7B). As cytoskeleton remodeling is a prerequisite step during the morphological transition of fibroblast-like preadipocytes into lipid-filled spherical mature adipocytes, we measured cytoskeleton proteins like beta-tubulin, beta-actin, and vimentin[128–130]. Beta-tubulin and beta-actin expression was significantly suppressed while vimentin levels were elevated in mice lacking VAT resident macrophages and mice deficient in myeloid SerpinB2 (Fig. 7C and Supplementary Fig. 7C). The absence of myeloid *Ifngr1* and IL-4 replenishment in *LysM*^cre/+ *SerpinB2*^fl/fl mice inflated the expression of beta-tubulin and beta-actin and diminished

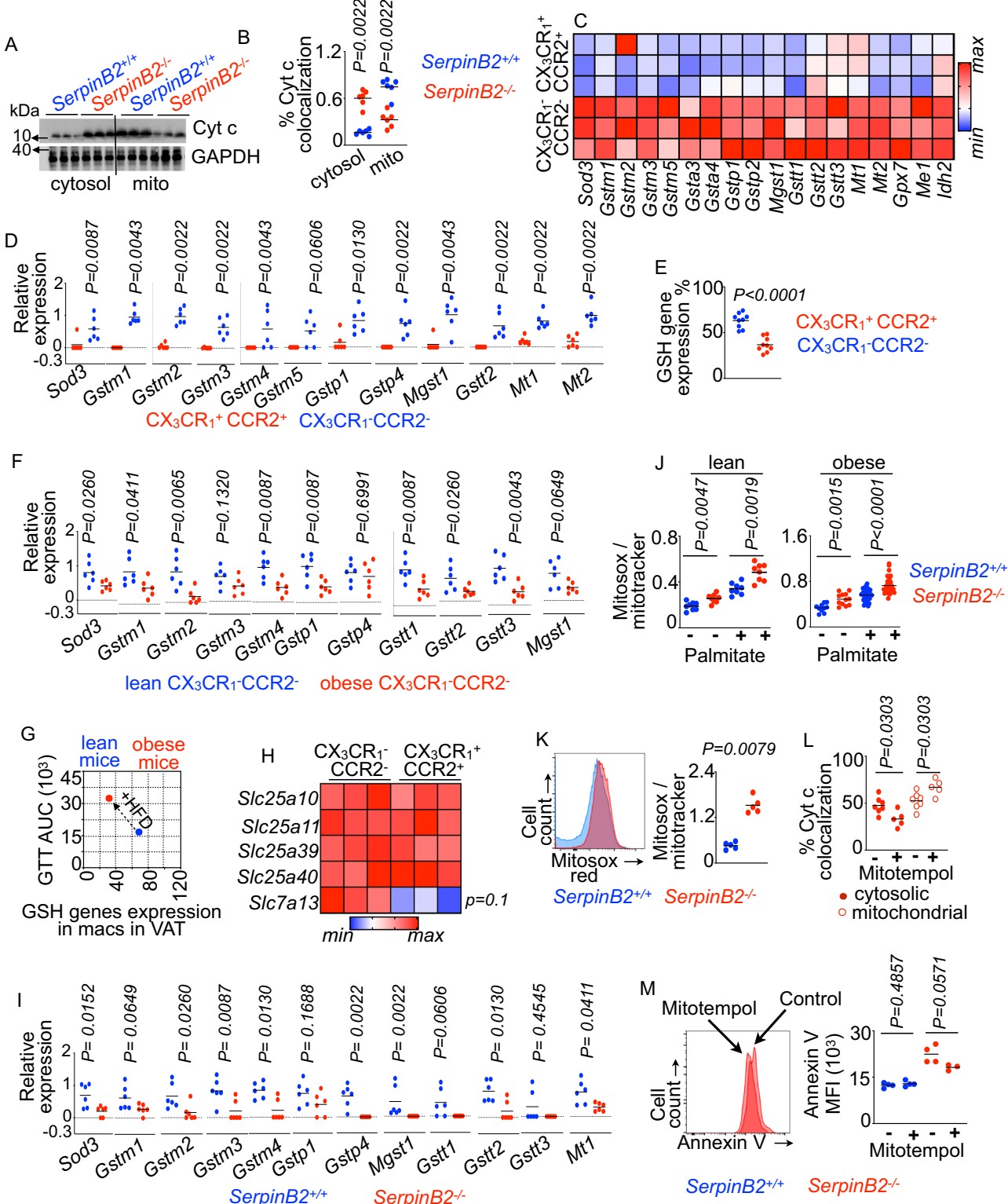

vimentin levels (Supplementary Fig. 7D, E). Supporting these observations, adipocytes residing near CCR2⁻ VAT resident macrophages were smaller (Fig. 7D).

**Glutathione (GSH) negatively regulates adipocyte hypertrophy and expansion of VAT in diet-induced obesity**

Given that CX₃CR1⁻ CCR2⁻ VAT macrophages express high levels of GSH, SerpinB2 deficiency lowers these antioxidants in VAT macrophages, and depletion of VAT resident macrophages increased

adipocyte size, we hypothesized that the anti-inflammatory and suppressed adiposity properties are attributable to GSH. N-acetylcysteine (NAC), an aminothiol and synthetic precursor of intracellular cysteine and GSH, dose-dependently abrogated the differentiation of adipocytes (Fig. 7E and Supplementary Fig. 7F). Concordant with the inhibitory effect on adipocyte development, NAC treatment promoted the expression of insulin sensitivity-promoting genes and blunted pro-inflammatory and adipose expansion genes in VAT depots (Fig. 7F). These results imply the

**Fig. 5 | Increased mitochondrial ROS in *SerpinB2*⁻/⁻ macrophages induces apoptosis. A, B** Cytochrome c (Cyt C) levels in the mitochondria and cytoplasm measured by immunoblot in *SerpinB2*⁻/⁻ or *SerpinB2*⁺/⁺ BMDM (n = 6/group). **C, D** Antioxidant gene expression using bulk RNA sequencing (**C**) (n = 3/group) and qPCR (**D**) (n = 6/group, each dot represents one mouse.) in the VAT macrophage subsets of lean mice is shown. **E** The bar graph represents the frequency of the VAT macrophage subsets expressing the GSH-encoding genes shown in (**C, D**) (n = 9/group). **F** Antioxidant gene expression using qPCR in CX3CR1⁻ CCR2⁻ VAT macrophages isolated from obese mice (n = 6/group). **G** The PCA plot shows the relation between GTT and GSH gene expression in macrophages of lean and obese mice. **H** Heat map showing the expression of the genes responsible for transporting GSH between cytoplasm and mitochondria (n = 3/group). **I** Antioxidant gene expression using qPCR in total macrophages sorted from obese *SerpinB2*⁺/⁺ and *SerpinB2*⁻/⁻ mice (n = 6/group). **J, K** Mitochondrial ROS was measured in untreated and palmitate-treated *SerpinB2*⁺/⁺ and *SerpinB2*⁻/⁻ BMDM isolated from lean mice and obese mice by confocal microscopy (**J**) (n = 8/group) and flow cytometry (**K**) (n = 5/group). **L, M** Cytosolic and mitochondrial Cytochrome c quantification in *SerpinB2*⁻/⁻ BMDM by immunofluorescence microscopy (**L**) (n = 7 for the without mitotempol group and 5 for with mitotempol group, each dot represents cells cultured in one well.) and assessment of annexin V MFI in *SerpinB2*⁻/⁻ or *SerpinB2*⁺/⁺ BMDM by flow cytometry (**M**) (n = 4 /group, each dot represents one mouse.) after mitotempol treatment. Mean ± s.e.m. * $P < 0.05$, ** $P < 0.01$, ***$P < 0.001$. The Mann–Whitney test (two-tailed) was used to determine the significance between two groups.

significant role played by GSH in confining the expansion of adipocytes in obesity. To ascertain the protective role of GSH, obese *LysM*^cre/+ *SerpinB2*^fl/fl mice were supplemented with 1% of NAC for 8 weeks in drinking water. NAC supplementation significantly curtailed fasting glucose levels, and improved glucose tolerance and insulin sensitivity compared to vehicle treatment (Fig. 7G and Supplementary Fig. 7G). Interestingly, NAC supplementation significantly reduced the body weight (Supplementary Fig. 7G). Moreover, insulin, triglycerides, free fatty acids, and free glycerol levels were reduced in the serum of NAC-treated obese *LysM*^cre/+ *SerpinB2*^fl/fl mice (Fig. 7G). Further, Akt activation was enhanced in muscle and VAT of NAC-supplemented *LysM*^cre/+ *SerpinB2*^fl/fl mice (Fig. 7H and Supplementary Fig. 7H). Consistently, Glut4 and adiponectin protein levels (Fig. 7H and Supplementary Fig. 7H) and insulin sensitivity-promoting gene expression (Fig. 7I) raised while the transcripts of inflammatory, adipose expansion, and insulin resistance-promoting genes were suppressed (Fig. 7I) in VAT of NAC-treated *LysM*^cre/+ *SerpinB2*^fl/fl mice. Furthermore, NAC supplementation escalated the number of tissue-resident macrophages in VAT (Fig. 7J and Supplementary Fig. 7I), reduced macrophage apoptosis (Supplementary Fig. 7J), and limited VAT weight (Supplementary Fig. 7K) and adipocyte size (Supplementary Fig. 7L). These results imply the significant role played by GSH in confining the expansion of adipocytes in obesity. Moreover, adipocytes cultured in the presence of conditioned media from *LysM*^cre/+ *SerpinB2*^fl/fl BMDM expanded more compared to *LysM*^+/+ *SerpinB2*^fl/fl BMDM (Supplementary Fig. 7M, N). However, adipocyte expansion was significantly attenuated with conditioned medium of NAC-supplemented *LysM*^cre/+ *SerpinB2*^fl/fl BMDM. In cluster, these findings endorse that the pro-inflammatory and glucose intolerance phenotype in the absence of myeloid *SerpinB2* could be attributed to the lack of GSH.

## Discussion

Tissue-resident macrophages are important in a myriad of physiological functions including organ regeneration[2–4], removal of cellular debris[5], injury[6], infection[7], nerve regeneration[4], neuronal function[12–15], mitochondrial homeostasis[9], electrical conduction[10], and protection against fibrosis[131–134]. In this study, we have shown that VAT resident macrophages are critical for maintaining an anti-inflammatory environment, VAT homeostasis, and insulin sensitivity. Tissue-resident macrophages mainly self-renew by local proliferation[135] and disappear during inflammation[7,10,44,48,136]. The mechanisms of the disappearance of tissue-resident macrophages in inflammation are not well understood. We observed that SerpinB2-mediated inhibition of cytochrome c release from the mitochondria is vital to adipose resident macrophage survival. Although other pathways, such as inflammasome signaling, could lead to macrophage apoptosis, the present study has not examined the contributions of other pathways in adipose resident macrophage depletion in obesity.

We observed that, among all leukocytes in VAT, resident macrophages expressed high levels of SerpinB2. The exact function of SerpinB2 in adipose resident macrophages is unknown. SerpinB2

deficiency precipitated apoptosis in this macrophage subset in obese, but not lean, mice indicating that SerpinB2 expression by adipose macrophages in the steady state is dispensable for their survival. Although the present study does not elucidate why SerpinB2 is required for VAT resident macrophage survival in obesity but not in the steady state, inflammation after initiation of an HFD may be a determinant of macrophage apoptosis in the absence of SerpinB2. Concordant with this hypothesis, we observed that IFN-γ was important in SerpinB2 reduction in obesity, and the anti-apoptotic role of SerpinB2 has been demonstrated in inflammatory disease pathologies[137–146]. Several mechanisms have been proposed for IFN-γ production by obese adipose tissue. In obesity, adipocytes secrete CXCL12, which recruits natural killer cells and stimulates the secretion of IFN-γ by these cells[147]. Besides natural killer cells, inflammatory macrophages and T helper cells can infiltrate adipose tissue and secrete IFN-γ in obesity[20,148]. Additionally, the activation of HIF-1α in response to the hypoxic environment in adipose tissue in obesity can stimulate the transcription of various genes including *Ifng*[149]. The significance of SerpinB2 expression by the macrophage subsets is not clearly understood. IFN-γ, which is released in high quantity in infection, decreases SerpinB2 levels in VAT resident macrophages, resulting in their apoptosis. Loss of VAT resident macrophages can lead to infiltration of monocyte-derived macrophages, which have better infection fighting ability and lower SerpinB2 expression, thereby are resistant to IFN-γ-mediated apoptosis. Although our data indicate that myeloid-specific deficiency of *Infgr1* resulted in improved insulin sensitivity by protecting adipose-resident macrophages from apoptosis, we cannot rule out the contributions of reduced food intake, increased activity, and reduced fat mass to the beneficial effects Future studies will be required to examine the significance of SerpinB2 expression by the macrophage subsets in infection clearance. Further, we found that SerpinB2-expressing CX₃CR1⁻ CCR2⁻ VAT macrophages are enriched with GSH-encoding genes. Antioxidants like GSH present in the cytoplasm modulate apoptosis by up-regulating anti-apoptotic proteins such as B-cell lymphoma 2[150,151]. An elevated ROS content in mitochondria is a driving factor of cytochrome c translocation from mitochondria to the cytoplasm via cardiolipin oxidation[152]. Surprisingly, SLC7A13, which mediates the transport of glutamic acid, one of the precursors for GSH synthesis[153,154] is expressed in higher amounts in CX₃CR1⁻ CCR2⁻ VAT macrophages. In line with this finding, the deficiency of SerpinB2, which is highly expressed by this subset of macrophages, attenuated GSH levels. As such, these observations indicate that SerpinB2-mediated increase in GSH levels curbs mitochondrial ROS production, which thwarts cytochrome c release in the cytoplasm, promoting macrophage survival. Whether SerpinB2 amplifies the level of GSH, a mitochondrial antioxidant, via SLC7A13 should be examined using mechanistic experiments.

Inflammatory cytokine production and immune cell function are controlled by mitochondrial respiration[155,156]. Recent reports demonstrated that mitochondrial respiration impairment by genetically depleting *Tfam*, which confers mitochondrial DNA stability, suppresses inflammation in macrophages[157–159]. Consistent with this

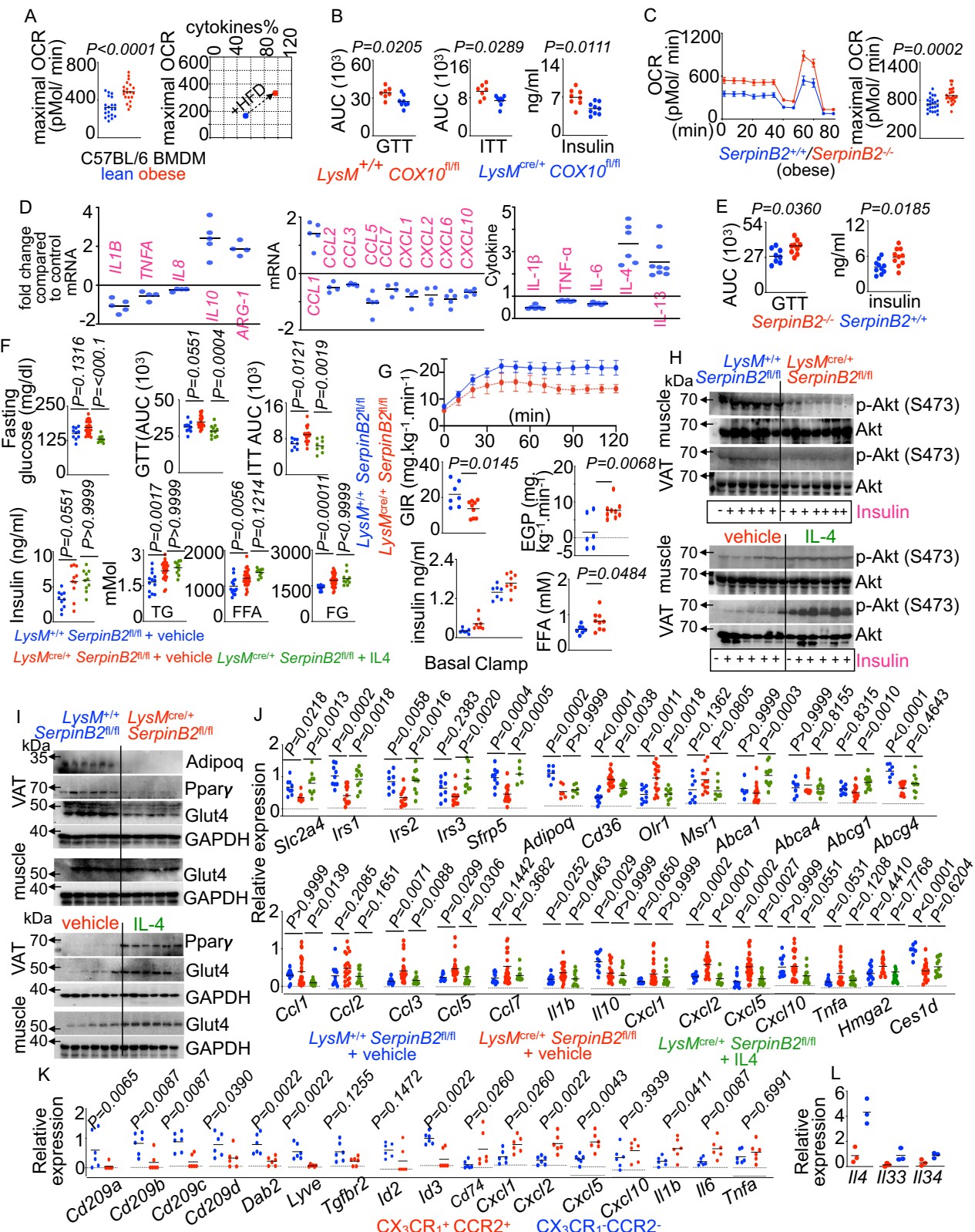

finding, we observed that the deficiency of myeloid *Cox10*, which encodes the mitochondrial complex IV[121-125], decreased inflammation, promoted VAT resident macrophage survival, and improved glucose tolerance. On the contrary, SerpinB2 depletion in macrophages kindled pro-inflammatory cytokine production with concomitant high oxidative phosphorylation and apoptosis of VAT resident macrophages, indicating the role of SerpinB2 in controlling exaggerated mitochondrial respiration, inflammation, and VAT resident macrophage survival in obesity. In line with these findings, we observed positive correlations among OXPHOS activity, inflammatory cytokines production, glucose intolerance, and insulin resistance in wild-type and myeloid-specific *SerpinB2* and *Cox10*-depleted mice. Altogether, these data strongly suggest the contribution of mitochondrial respiration in inflammation and VAT macrophage survival. However,

**Fig. 6 | SerpinB2 deficiency in macrophages exacerbates inflammation.**
**A** Oxygen consumption rate (OCR) was measured by Seahorse XF 96 Extracellular Flux Analyzer (n = 20/group) in BMDM isolated from lean and obese mice. The correlation between maximal OCR and cytokine levels measured by ELISA in BMDM supernatants is shown. **B** GTT, ITT, and fasting insulin levels in HFD-fed *LysM*<sup>+/+</sup> *COX10*<sup>fl/fl</sup> and *LysM*<sup>cre/+</sup> *COX10*<sup>fl/fl</sup> mice (n = 7 for WT and 8 for KO, combined data of two independent experiments). **C** The Seahorse traces and maximal OCR (n = 24/group) in BMDM of obese *SerpinB2*<sup>−/−</sup> or *SerpinB2*<sup>+/+</sup> mice are shown. **D** mRNA and protein levels of pro- and anti-inflammatory cytokines by qPCR and ELISA, respectively, were determined after *SerpinB2* overexpression in THP-1 macrophages (n = 4/group, each dot represents cells cultured in one well.). **E** GTT and fasting insulin levels (n = 6–8/group, combined data of two independent experiments) (n = 8 for WT and 9 for KO) in obese *SerpinB2*<sup>+/+</sup> and *SerpinB2*<sup>−/−</sup> mice. **F–L** HFD-fed *LysM*<sup>+/+</sup> *SerpinB2*<sup>fl/fl</sup> and *LysM*<sup>cre/+</sup> *SerpinB2*<sup>fl/fl</sup> mice were injected with PBS-vehicle, and *LysM*<sup>cre/+</sup> *SerpinB2*<sup>fl/fl</sup> mice were injected with IL-4. **F** GTT and ITT were performed, and the concentrations of fasting blood glucose, serum insulin,

triglycerides (TG), free fatty acids (FFA), and free glycerol (FG) were evaluated (n = 10 for WT+vehicle, 18 for KO+vehicle, and 11 for KO + IL-4, combined data of at least two independent experiments). **G** Glucose infusion rate (GIR), hepatic endogenous glucose production (EGP), plasma insulin, and FFA levels are determined by hyperinsulinemic-euglycemic clamp studies (n = 7–9/group). Immunoblots (**H**, **I**) show pAkt, total Akt, Glut-4, adiponectin, and Pparγ expression in muscle and VAT (n = 5-6/group). The expression of the metabolic and inflammatory genes is measured by qPCR in whole VAT (**J**) (n = 8/group) and sorted VAT macrophages (**K**) (n = 6/group). **L** The expression of the anti-inflammatory genes was determined by RNA sequencing in CX<sub>3</sub>CR<sub>1</sub><sup>+</sup> CCR2<sup>+</sup> and CX<sub>3</sub>CR<sub>1</sub><sup>−</sup> CCR2<sup>−</sup> VAT macrophages isolated from lean mice (n = 3/group). Mean ± s.e.m. *P < 0.05, **P < 0.01, ***P < 0.001. The Mann–Whitney test (two-tailed) was used to determine the significance between two groups. One-way ANOVA with Bonferoni's post hoc correction test was performed to determine differences among data obtained from more than two groups (Fig. 6F, J).

---

the mechanisms of how obesity reprograms mitochondrial respiratory capacity in macrophages, which in turn rewires inflammatory phenotype, need to be explored.

Obesity induces adipocyte hypertrophy, monocyte infiltration into adipose tissue, and their subsequent differentiation into macrophages[30,87,160–162]. How adipose macrophages regulate pathogenic adipocyte hypertrophy is not known. Our study reveals that adipose resident macrophages control adipocyte expansion, accentuating their protective effect in obesity, which is in line with a recent study showing the control of lipid stores by resident macrophages[163]. Adipocyte expansion is regulated by several biological processes including adipocyte apoptosis and necrosis, adipogenesis, lipolysis, and lipogenesis. Inflammatory cytokines produced by macrophages and monocytes can regulate these processes. A recent study showed that PDDGcc produced by macrophages can control adipocyte energy storage, which aids in the growth of adipose tissue in obesity[163]. Adipose resident macrophages express low levels of *Il1b*, which stimulates adipocyte growth[126], and high levels of *Tgfb2*, which inhibits adipocyte growth[127], compared to monocyte-derived macrophages. However, how these inflammatory cues regulate adipocyte growth by affecting adipocyte apoptosis and necrosis, adipogenesis, lipolysis, or lipogenesis is beyond the scope of this study.

In summary, our findings imply a significant role of resident macrophages present in VAT to protect against metabolic complications. While we acknowledge that the data in Fig. 1 establish a correlation between VAT resident macrophage loss and emergence of insulin resistance in obesity, the experimental depletion of this macrophage subset highlights their importance in maintaining insulin sensitivity. Hence, the strategies to prevent VAT resident macrophage loss may attenuate VAT expansion and improve insulin sensitivity in obesity. Moreover, although VAT resident macrophages express SerpinB2 abundantly, other tissue-resident macrophages important in metabolic syndromes, such as Kupffer cells, may express this protein. However, Kupffer cell numbers in the liver did not change in *SerpinB2*<sup>−/−</sup> obese mice. The mechanisms of the absence of numerical changes in Kupffer cells of these mice should be elucidated by future studies. Another limitation of this study is that we used samples from deceased patients, which may introduce many potential confounders. These patients had highly variable BMI, and the information regarding their causes of death was not available. Furthermore, to investigate the importance of GSH in the regulation of VAT physiology and systemic insulin sensitivity, we employed a pharmacological approach. This experiment does not specifically test the significance of macrophage GSH. Although our experiments show that secreted factors from macrophages treated with NAC restrain adipocyte expansion in culture, macrophage-specific GSH transgenic mice will be useful for this purpose. Furthermore, we do not know what factor(s) is

secreted by macrophages in the presence of NAC can limit adipocyte size. Additionally, the beneficial effects of NAC may be partially attributed to the reduction in body weight observed after treatment. Finally, future studies addressing the effects of VAT resident macrophages on mesenchymal stem cell differentiation into adipocyte precursors increasing adiposity should be performed.

## Methods
### Mouse strains, cellular fate mapping, and treatments
The control and experimental mice were co-housed in a barrier facility at the University of Pittsburgh. Both male and female mice (16–50 weeks old) wildtype and transgenic mice have been used for the experiments. To investigate the origin of adipose tissue macrophages, female ROSA-tdTomato mice were mated with male *Cx₃cr1*<sup>creER</sup> mice. We injected tamoxifen (50 μl of 20 mg/ml solution, Sigma, T5648) dissolved in corn oil (Sigma, C8267) in the offspring to induce permanent tdTomato expression in Cx₃cr1<sup>+</sup> macrophages. *Cx₃cr1*<sup>GFP/GFP</sup>, *LysM*<sup>cre/cre</sup>, *Ifngr*<sup>fl/fl</sup> (Jackson Lab, #025394), and *SerpinB2*<sup>−/−</sup> (Jackson Lab, #007234) mice were purchased. *SerpinB2* Tm1c mice (KOMP project ID CSD66707) were bred with Flp recombinase-expressing mice (Jackson Lab, #009086) to generate conditional-ready Tm1a mice. Tm1a mice were crossed with wild-type female mice purchased from the Jackson Lab at least for five generations. After this, Tm1a mice were bred with *LysM*<sup>cre/cre</sup> mice to generate *LysM*<sup>cre/+</sup>/*LysM*<sup>+/+</sup> *SerpinB2* <sup>fl/fl</sup> mice. Eight to twelve weeks old male and female mice were fed with either a chow or HFD (Research Diets Inc, #D12492) for four months. Ten weeks old *LysM*<sup>cre/+</sup>*SerpinB2* <sup>fl/fl</sup> mice were fed with an HFD for two months followed by either an i.p infusion of 2 doses/week of 1 ng IL-4 (R&D, #404-ML-050/CF) dissolved in 100 microliters of PBS or supplementation of 1% NAC (Research Products, #A10040) dissolved in water for 8 weeks as previously[164]. The mice were randomized among various groups, and most analyses were blindfolded. The mice were euthanized using carbon dioxide. The sources and identifiers of all reagents and resources used in this study have been included in Supplementary Table 2.

### Study approval
All animal experimental protocols were approved by the Institutional Animal Care and Use Committee of the University of Pittsburgh. Omental adipose tissue from patients with diabetes and normal individuals was collected via a warm autopsy program directed by Dr. Mauricio Rojas. His team obtained consents from decedents' next-of-kin or legal representatives. These patients were classified as either lean (<25 BMI) or obese (>30 BMI) based on their BMI. Collection of the tissue samples from deceased patients was approved by the University of Pittsburgh Committee for Oversight of Research and Clinical Training Involving Decedents (CORID #724).

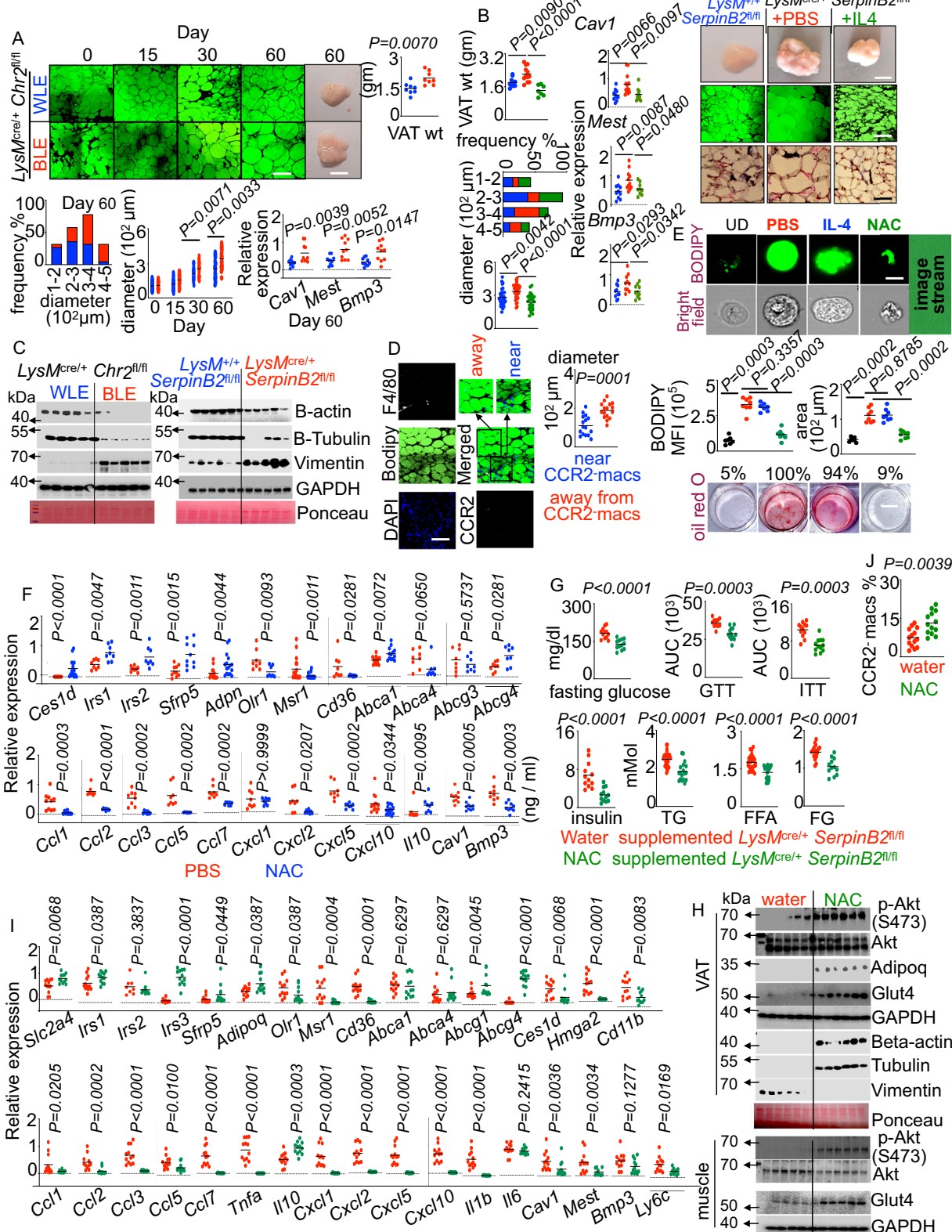

## Parabiosis

CD45.1 (Jackson Lab, #002014) and CD45.2 (Jackson Lab, #000664) mice were surgically joined. Following anesthesia and disinfection using betadine scrub and 70% isopropanol, an incision was made from the ear to the base of the tail on the adjacent sides of the mice. The subcutaneous fascia was loosened, and the skin was sutured using a 5–0 silk suture. The olecranon and knee joints were attached to avoid stress on the suture lines. Non-absorbable sutures were removed on day 7 after the surgery, and chimerism was ensured on day 21 after parabiosis by flow cytometric analyses of peripheral blood leukocytes.

## Intravital microscopy

tdTomato expression was induced in CX₃CR1-expressing cells by injecting 100 µl of 20 mg/ml of tamoxifen (Sigma) dissolved in corn oil

**Fig. 7 | Glutathione (GSH) restrains adipocyte hypertrophy and VAT expansion in diet-induced obesity. A, B** BODIPY images, adipocyte diameters and frequencies, and the expression of VAT expansion genes are shown in obese *LysM*^cre/+^/*Chr2*^fl/fl^ mice exposed to either white (WLE) or blue (BLE) light (n = 8/group), and obese *LysM*^+/+^ *SerpinB2*^fl/fl^ and *LysM*^cre/+^ *SerpinB2*^fl/fl^ mice treated with vehicle and *LysM*^cre/+^ *SerpinB2*^fl/fl^ mice treated with IL-4 (n = 7/group). Scale bar=20 μm for BODIPY and bright field images, and 200 μm for VAT images. **C** Immunoblot images of the proteins in VAT of obese *LysM*^cre/+^/*Chr2*^fl/fl^, *LysM*^+/+^ *SerpinB2*^fl/fl^, and *LysM*^cre/+^ *SerpinB2*^fl/fl^ mice (n = 5–6/group). **D** Confocal imaging to show the association between CCR2− macrophages and adipocyte size in mouse VAT (n = 16/group). Scale bar =20 μm. **E** BODIPY and bright field images using ImageStreamX Mark II and Oil Red O pictures of undifferentiated or differentiated 3T3L1 cells cultured in the presence or absence of IL-4 or NAC (n = 7/group). Scale bar = 5 μm for the ImageStreamX images and 200 μm for the Oil Red O pictures. **F** The metabolic and inflammatory genes were measured by qPCR in 3T3 L1 differentiated adipocytes exposed to vehicle or NAC (n = 8/group). **G–J** Obese *LysM*^cre/+^ *SerpinB2*^fl/fl^ mice were supplemented with either water or NAC-containing water. **G** GTT and ITT were performed, and the concentrations of fasting blood glucose, serum insulin, triglycerides (TG), free fatty acids (FFA), and free glycerol (FG) were evaluated. (n = 8/group). **H** Immunoblots showing pAkt, total Akt, Glut4, adiponectin, and Ppar*γ* expression in muscle and VAT (n = 6/group). **I** The expression of the metabolic and inflammatory genes was analyzed in VAT by qPCR. (n = 12/group). **J** Quantification of CCR2− macrophages in VAT. (n = 14/group). **I, J** Each dot represents one mouse. Mean ± s.e.m. * P < 0.05, ** P < 0.01, ***P < 0.001. The Mann Whitney test (two-tailed) was used to determine the significance between two groups. One-way ANOVA with Bonferoni's post hoc correction test was performed to determine differences among data obtained from more than two groups (Fig. 7B). The Mann–Whitney test (two-tailed) was used to determine the significance between two groups.

(Sigma) once a day for five consecutive days into CX$_3$CR1^CreER/+^ ROSA^tdTomato/+^ mice. By using betadine and 70% ethanol, the abdominal skin was disinfected, and an incision was made on the linea alba. Gonadal adipose tissue was retracted and placed in sterile 0.9% saline. Gonadal adipose tissue was imaged for 30–60 minutes per mouse by using a two-photon microscope that was equipped with a heated mouse holder to maintain a body temperature of 37 °C. To maintain contact between the tissue and the microscope objective, Methocel 2% gel was used during imaging. The total imaging duration was 30–60 minutes per mouse. Data analysis was performed using ImageJ.

## Gamma irradiation and bone marrow transplantation
Non-anesthetized mice were placed in a gamma irradiator and exposed to 10 Gy irradiation. Then the mice were returned to the housing facility and injected with one million bone marrow cells/mouse i.v under anesthesia.

## Organ harvesting, flow cytometry, and cell sorting
Mice were euthanized and perfused thoroughly with 30 mL of ice-cold PBS through the left ventricle after incising the right atrium to remove circulating blood from solid organs such as adipose tissue. Gonadal and mesenteric adipose tissues were harvested and digested in PBS containing 1 mg/ml of collagenase IV (Worthington, # LS004209) and 20 μM HEPES buffer at 37 °C on thermoblock by rotating at 750 rpm for 30 minutes. After red blood cell lysis for three minutes using RBC lysis buffer (BioLegend, #420302), blood cells were washed and resuspended in cold FACS buffer (PBS containing 0.5% BSA). Cell staining and analysis were performed as described previously[44,165]. All antibodies used in this study were purchased from eBioscience, BioLegend, and BD Biosciences. Anti-mouse and human CCR2 antibodies were used in 1:30 dilutions, and all other antibodies were used in 1:600 dilutions in FACS buffer before flow cytometry acquisition. We used these following antibodies: anti-CD11b (BD Biosciences, M1/70 # 557657), CD11c (BioLegend, N418 # 117338, BD Biosciences, HL3 #553800), Ly6G (BD Biosciences, 1A8 # 563979), CD115 (eBioscience, AFS98 # 46-1152-82), Ly-6C (BioLegend, HK 1.4#128006), CD19 (BD Biosciences, 1D3 # 563148), MHC class II (BioLegend, M5/114.15.2 # 107620), CD64 (BD Biosciences, X54-5/7.1 # 558455), F4/80 (BioLegend, BM8#123114), CD45.1 (BioLegend, A20 #110730), CD45.2 (BioLegend,104 # 109820, BD Biosciences, 104 # 560693), CCR2 (R&D Systems, # FAB5538A) and streptavidin (BD Biosciences, #563260, 563261). Neutrophils and monocytes were gated as CD11b+ Ly6G+ and CD11b+ CD115+, respectively. CD45+ CD11b+ CD64+ F4/80+ cells in adipose tissue were considered as macrophages. Human omental leukocytes were extracted using the same digestion method mentioned above and stained with antibodies against CD45 (BD Biosciences, HI30 # 564585), CCR2 (BioLegend, K036C2 #357206), CD24 (BD Biosciences, ML5 # 561647), CD14 (BD Biosciences, #555399), CD16 (BD Biosciences, 3G8# 560195), CD11c (BD Biosciences, B-ly6 # 563404), CD 206 (BD Biosciences, 19.2 # 564063), and HLA-DR (BD Biosciences, G46-6 # 565127). CD45+ CD11c+ CD206+ cells in adipose tissue obtained from patients were deemed as macrophages. Fortessa Flow Cytometers at the Unified Flow Cytometry Core at the University of Pittsburgh were used to perform data acquisition. Data were analyzed using FlowJo software (Tree Star). To assess the apoptosis of adipose tissue macrophages, the cells were first stained with antibodies mentioned above. FITC Annexin V apoptosis detection I (BD Biosciences, #556547) and PE-active Caspase 3 apoptosis (BD Biosciences, #550914) kits were used according to the manufacturer's instructions.

## Whole-mount imaging of VAT using confocal microscopy
Following collection of VAT from patients and mice, the samples were fixed with 4% formalin for 2 hours and stored in 30% sucrose solution containing 0.05% sodium azide overnight before storing at −80 °C. Small pieces of formalin-fixed VAT were cut, permeabilized with 0.1% triton X-100 for 4 hours and blocked overnight in PBS containing 2% BSA in 96 well cell culture plates. Tissues were incubated for 48 hours with the primary antibodies against proteins like F4/80 (Invitrogen, #MAI-91124), cleaved caspase 3 (Abcam, #ab13847), CD11b (Abcam, #ab133357), CX$_3$CR1 (Abcam, #ab8021), CCR2 (Bio-Rad, #AAM72), Ki67[81,91,129], CD68 (ThermoFisher Scientifics, # 14-0688-82), and SerpinB2 (Invitrogen, #PA5-27857) followed by washing with PBS and incubation for 24 hours at 4 °C with Alexa fluor 488, 594 and 647-conjugated secondary antibodies. Tissue sections were stained and mounted with Vectashield DAPI (Vector Laboratories, #H-1200). Nikon confocal laser microscopy was used to obtain images of the stained tissue sections. We used Image J and NIS-Elements to discern the frequency and number of cell populations and calculate the mean fluorescence intensities of proteins. For measuring adipocyte size, small pieces of VAT were stained with 1 μg/ml BODIPY (ThermoFisher Scientifics, # D3922) for 15–30 minutes followed by PBS washing in 96-well plates and mounting with Vectashield DAPI. About 10–15 images were randomly selected for every sample, 100–200 adipocytes were selected randomly from every image, and adipocyte size was measured using ImageJ.

## Glucose and insulin tolerance tests
Two-month-old mice that were fed with a 60% calorie-rich HFD for 120 days were fasted for 6 hours before glucose and insulin tolerance tests, Blood was collected from these mice by tail nicking to measure glucose concentrations. Fasting blood glucose concentrations were measured and considered as a 0-minute time point. Sterile glucose solution (1 g/kg body weight dissolved in PBS to make a 10% solution) was injected intraperitoneally, and glucose concentrations were measured at various time points. For the insulin tolerance test, mice were injected with insulin (0.25 U/kg body weight, Fisher Scientifics, #12-585-014) and blood glucose levels were measured at 15, 30, 60, 90, and 120 minutes after injection.

### Assays to measure serum concentrations of insulin

Mice were anesthetized using 2% isoflurane mixed with oxygen. Blood was collected via retro-orbital bleeding using non-heparin-coated capillary tubes and allowed to clot. Samples were centrifuged (3000 rpm for 10 minutes at 4 °C) to separate serum, which was stored at −80 °C until further analyses. An ELISA-based assay was performed to measure serum insulin concentrations (Mercordia Diagnostics, #10-1247-01).

### Estimation of serum triglycerides, free fatty acids, and free glycerol concentrations

Following serum separation, triglycerides, free glycerol, and free fatty acids were quantified using the following reagents or kits: triglycerides liquid reagent (Pointe Scientific, #T7532120), EnzyChrom free glycerol assay kit™ (Bioassay Systems, #EGLY-200), and EnzyChrom free fatty acid assay kit™ (Bioassay Systems, #EFFA-100).

### Cell sorting and quantitative RT-PCR

We obtained single-cell suspensions of adipose tissue as described above. The cells were stained for macrophage markers. VAT monocyte-derived and resident macrophages were sorted using a FACS Aria into RNAse-free tubes containing 100 μl of RNA extraction buffer. mRNA was extracted using a commercially available kit (Applied Biosystems, #12204-01) and cDNA was made by a reverse transcription assay using high-capacity cDNA kit (ThermoFisher Scientific, #4368813). Genes encoding cytokines were quantified with a Powerup™ SYBR™ Green mastermix (Applied Biosystems™, #A25778)-based quantitative RT-PCR assay (Applied Biosystems). Ct values were normalized to the housekeeping gene *Gapdh* and *18sRNA*, and gene expression was ascertained. Excel and Prism were used to generate heatmaps based on the gene expression values. The primer sequences have been provided in Supplementary Table 3.

### Cell sorting and RNA-seq analysis

We obtained single-cell suspensions of adipose tissue as described above. The cells were stained for macrophage markers. VAT monocyte-derived and resident macrophages were sorted using a FACS Aria into RNAse-free tubes containing 100 μl of RNA extraction buffer. mRNA was extracted using a commercially available PicoPure™ RNA isolation kit (Applied Biosystems, #12204-01). mRNA sequencing was performed by the Health Science Sequencing Core at UPMC Children's Hospital of Pittsburgh and FASTQ files have been deposited to GSE 118226[44]. Using CLC Genomics Workbench 21, RNA seq data analysis was performed, and a QC report was generated to check the quality of the data. The adaptor sequences (CTGTCTTATA) were trimmed from the 3′ end of the reads. The trimmed reads were then mapped against the *Mus musculus* CLC reference genome version 86. Gene expression was normalized as reads per kilobase of transcript, per million mapped reads (RPKM), and the genes that met a cut-off of FDR $P < 0.05$ and 2 log2 fold change in $CX_3CR1^+ CCR2^+$ compared to $CX_3CR1^- CCR2^-$ adipose macrophages were calculated using the software. A Volcano plot to depict differential gene expression was made, and a heatmap showing the genes having at least 2-fold differences between the macrophage subsets and less than FDR $P$ value of 0.05 was generated using CLC Genomics Workbench. Next, we used Ingenuity Pathway Analysis to find canonical and disease pathways differentially expressed between the two subsets of macrophages.

### Gene expression analysis using quantitative RT-PCR

Following collection, organs were stored overnight in RNA later (Sigma, #R0901) and transferred to −80°C after removing RNA later. mRNA was extracted using an RNEasy kit (Qiagen, # 74104), and cDNA was made by a reverse transcription assay. The genes encoding cytokines were quantified with a SYBR Green-based quantitative RT-PCR assay (Applied Biosystems). Ct values were normalized to the housekeeping genes *Gapdh* and 18S RNA, and gene expression was ascertained. Prism and Excel were used to generate heat maps based on the gene expression values.

### Apoptosis induction in VAT resident macrophages

Single-cell suspensions of VAT were obtained using collagenase digestion as written above. The cells were cultured in 1 ml DMEM containing 20 μM camptothecin (Sigma, #C9911) for 2 hours to induce apoptosis. The cells were stained with antibodies against leukocyte markers. The apoptosis kits were used to quantify apoptotic macrophages by flow cytometry. For annexin V staining (BD Biosciences, #556547), the cells were suspended in 100 μl of 1x binding buffer and stained with 5 μl annexin V Alexa Fluor 488 and 1 μl of 100 μg/ml propidium iodide (BioLegend, #421301). The cells were incubated at room temperature for 15 minutes, and 400 μl of 1x binding buffer was added and analyzed by flow cytometry immediately. For caspase 3 staining (BD Biosciences, #550914), the cells were incubated in 500 μl of BD cytofix/cytoperm buffer for 20 minutes on ice and were centrifuged at 370 RPM for 7 minutes at 4 °C followed by washing in 500 μl of the perm/wash buffer. The cells were then incubated in 100 μl of perm/buffer containing 5 μl of caspase 3 antibody for 30 minutes at room temperature. The cells were washed in 1 ml of perm/wash buffer and diluted in 400 μl of perm/wash buffer before flow cytometry analysis.

### Insulin stimulation in the vena cava

Two months old mice $LysM^{+/+}$ $SerpinB2^{fl/fl}$ and $LysM^{cre/+}$ $SerpinB2^{fl/fl}$, $LysM^{+/+}$ $Ifngr1^{fl/fl}$ and $LysM^{cre/+}$ $Ifngr1^{fl/fl}$ mice were fed on 60% calorie-rich high-fat diet for 16 weeks. These mice were fasted overnight for 16 hours and anesthetized with isoflurane and injected with or without insulin (0.25 U/kg body weight, Fisher Scientifics, #12-585-014). Fifteen minutes after insulin injection, liver, muscle, and visceral adipose tissue were harvested. These tissues were processed for immunoblot to check the activation status of Akt using antibodies that detect total and phosphorylated Akt.

### BMDM culture and treatments

Femurs and tibias were collected from $SerpinB2^{+/+}$, $SerpinB2^{-/-}$, $LysM^{+/+}$ $SerpinB2^{fl/fl}$, and $LysM^{cre/+}$ $SerpinB2^{fl/fl}$ mice after left ventricular perfusion with PBS to collect bone marrow cells, which were RBC lysed and plated in cell culture dishes. After six hours, floating cells were transferred into new dishes and these cells were used for BMDM culture. Adherent cells were cultured in DMEM containing 5.5 mM glucose and supplemented with 20% L929 conditioned medium for 6−8 days.

Similarly, BMDM were cultured in 5.5 mM glucose containing DMEM along with 20% L929 conditioned medium. BMDM obtained from the mice mentioned above were grown in 60 mm dishes for 2 days, and the levels of IL-1β (ThermoFisher Scientifics, #88-7013-88) and IL-6 (ThermoFisher Scientifics #88-7064-88) were measured using ELISA. Additionally, cells were collected, RNA was made using an RNEasy kit (Qiagen, #74104), and gene expression was quantified. Similarly, BMDM from bone marrow cells of C57BL/6 mice were cultured and treated with 250 milliunits of IFN-γ (Invitrogen, #PHC4031) or various chemicals such as palmitate (Acros Organics, #416700050), cycloheximide (Alfa Aesar, #J6690103), and 250 μM mitotempol (Fisher Scientific, #NC1229394) for 0-24 hours in either 6 well plates or chamber slides. In a separate experiment, BMDMs were treated with or without Ikaros inhibitor lenalidomide (5 μM) (MedChemExpress, #HYA0003) for 2 hours followed by stimulation with 250 milliunits of IFN-γ for 24 hours.

THP-1 cell culture: THP-1 monocytes were seeded at a density of $2 \times 10^5$/ml and differentiated into macrophages in the presence of 100 nM Phorbol-12-myristate-13-acetate (PMA) (Calbiochem, # 524400) for 2 days in RPMI low glucose media in 6-well plates or 4-well chambered slides as described previously[44,165]. Following differentiation, macrophages were cultured without any treatment for 2 more

days by changing medium for every 24 hours. Then the cells were treated with or without NF-kB inhibitor Bay-117082 (MedChemExpress, #HY-13453) for 2 hours followed by stimulation with 250 milliunits of IFN-γ for 24 hours. The cells were either fixed in 4% paraformaldehyde for immunofluorescence or used for gene expression analysis.

## 3T3L1 mouse fibroblasts culture and treatment

3T3- L1 fibroblasts were cultured in Zen Bio Inc 3T3-L1 preadipocyte Medium (#PM1L1) until they reached about 50% confluence in 6 well plates, followed by IL4 (10 ng/ml) or NAC treatment and differentiation with Zen Bio Inc adipocytes differentiation medium (#DM2L1). After two days, differentiated cells were maintained in Zen Bio Inc adipocytes maintenance medium (#OMAM) for 10 days with periodic changes of the culture medium, IL4, and NAC every two days as previously[166,167]. To delineate tissue-resident macrophages' effects on adipocyte development, two days after differentiation 3T3-L1 fibroblasts derived adipocytes were cultured in the presence of conditioned media of BMDM derived from $LysM^{+/+}$ $SerpinB2^{fl/fl}$, $LysM^{cre/+}$ $SerpinB2^{fl/fl}$ mice, and NAC-treated $LysM^{cre/+}$ $SerpinB2^{fl/fl}$ mice. The BMDM was cultured in the conditioned medium along with the maintenance medium (1:1 ratio) for 6–8 days. BODIPY staining and confocal microscopy were performed to assess adipocyte size.

## Analysis of mitochondrial volume, membrane potential, ROS, and cytochrome c in VAT macrophages and BMDM using flow cytometry and confocal microscopy

Single-cell suspensions from VAT were obtained using the method discussed above. Cells were incubated with the antibodies against Ly-6C, F4/80, and CD11b. Following staining with the antibodies, the cells were treated with 200 nM mitotracker green (Invitrogen, #M7514, Excitation/emission: 490/516, in FITC channel) to detect all mitochondria, 200 nM mitotracker deep Red (Invitrogen, #M22426, Excitation/emission: 644/665, in APC channel) to detect mitochondrial membrane potential, and 5 μM mitosox red (PE) (Invitrogen, #M36008, Excitation/emission: 644/665 in APC channel) to detect mitochondrial reactive oxygen species. For confocal microscopy, BMDM were treated with or without 250 μM palmitate for 24 hours and incubated with 200 nM mitotracker green, 200 nM mitotracker deep Red (Excitation/emission: 644/665, in the Cy3 channel), and 5 μM of mitosox red (PE), and counterstained with 300 nM DAPI (Invitrogen, #D1306) for 30 minutes in PBS. To quantify nuclear vs. cytoplasmic cytochrome c, we gently dropped 100 μl of cytochrome c antibody (Novus Biologicals, #SC69-08) solution into the wells containing BMDM and incubated overnight at 4°C. After washing with PBS containing 0.5% BSA (PBB) five times, the cells were incubated for 60 minutes with goat anti-mouse secondary antibody conjugated with Cy3. Images were taken using a Nikon 216.3 confocal microscope.

## Hyperinsulinemic euglycemic clamps studies

These were performed according to guidelines established by the Mouse Metabolic Phenotyping Center Consortium[168]. Briefly, an indwelling jugular vein catheter was surgically implanted five days before the clamp. Mice were fasted overnight prior to the study and received an infusion of [3-3H]-glucose (HPLC purified; Perkin-Elmer Life Sciences) at a rate of 0.05 μCi/min for 120 min followed by blood sampling via tail vein massage to determine basal (fasted) rates of glucose turnover. Next, mice received a primed/continuous infusion of insulin and [3-3H]-glucose (Novolin; Novo Nordisk; 10.5 mU/kg/min and 0.36 μCi/min prime for 4 min; 4.5 mU/kg/min and 0.1 μCi/min continuous) and a variable infusion of 20% dextrose to maintain euglycemia. Blood glucose was sampled at 10 min intervals for the 120 min infusion and plasma samples were collected every 10 min during the last 40 min of the clamp for determination of glucose enrichment for turnover calculations and plasma insulin levels.

## Immunoblot

Following euthanization and perfusion with PBS through the left ventricle of mice, the liver and quadriceps muscles were harvested and snap frozen in liquid $N_2$. Tissues and BMDM were homogenized in RIPA lysis buffer (Alfa Aesar, #J63324) containing phosphatase and protease (Pierce, #88666) inhibitors. Homogenates were subjected to SDS-PAGE, and blots were probed with primary antibodies against adiponectin (CST, #2789), beta-tubulin (CST, #2128), beta-actin (CST, #4970), vimentin (Sigma, #SAB4200716), glut4 (Fisher Scientifics, #MA183191), SerpinB2 (Invitrogen, #PA5-27857), Akt (CST, #4691), p-Akt ser 473 (CST, #4060), and GAPDH (CST, #2118). For measuring cytochrome c levels in the cytoplasm and mitochondria, cells were subjected to cytosol and mitochondrial fractionation using a cytosol/mitochondria extraction kit (Biovision, #K256-25) and SDS-PAGE was incubated with a cytochrome c antibody (Novus Biologicals, # SC69-08) for overnight at 4 °C. Subsequently, we incubated the blots with secondary IgG-HRP antibody, and images were captured by molecular imaging system as described previously[44,165].

## Cellular mitochondrial bioenergetics of BMDM

Bioenergetic analysis of oxidative phosphorylation was performed using Seahorse XF 96 Bioanalyzer. Cells were seeded in collagen type I–coated Seahorse plate and treated when confluent. Following treatment, cells were washed with warm media, and the growth medium was replaced with a bicarbonate-free medium. After attaining equilibration for 60 minutes in a 37 °C incubator without $CO_2$, cells were sequentially treated with 20 μM oligomycin, 20 μM FCCP at 45 minutes, and 20 μM rotenone at 62 minutes in the presence or absence of etomoxir, a carnitine palmitoyltransferase 1b (CPT1b) inhibitor. Baseline respiration, respiratory ATP production, and maximal respiration rates were measured and normalized to protein content by performing protein analysis.

## VAT resident macrophages depletion in $LysM^{cre/+}$ $ChR2^{fl/fl}$ mice

We generated $LysM^{cre/+}$ $ChR2^{fl/fl}$ mice that express the channelrhodopsin-2 (ChR2) protein in macrophages and induced apoptosis in VAT resident macrophages by exposing epididymal adipose tissue to the blue light (440 nm) for 20 minutes as earlier[44]. For control, epididymal adipose tissue was exposed to the white light for 20 minutes. To investigate the function of VAT resident macrophages in systemic glucose clearance and insulin sensitivity preservation in obesity, blue light-exposed (BLE) and white light-exposed (WLE) mice were fed on a high-fat diet for 15-60 days.

## Calculations of insulin sensitivity, insulin resistance, and GSH gene expression

We combined the expression of insulin sensitivity and anti-inflammatory genes obtained by RNA sequencing and qPCR in the VAT macrophage subsets and considered as 100%. Based on this combined gene expression, we calculated the expression of individual genes as percentages in Fig. 1J. Similarly, for the GSH-encoding genes, we represented the combined the expression of these genes obtained from the RNA sequencing and qPCR experiments as 100%. Based on this average expression, we calculated the expression of individual genes and then averaged them to represent as a single value in Fig. 5E.

## PCA Plots

The PCA plot for GTT (AUC) and GSH gene expression in $CX_3CR1^-$ $CCR2^-$ VAT macrophages was made using the calculated percentage expression of the GSH-encoding genes (Gstm1, Gstm2, Gstm3, Gstm4, Gstp1, Gstp4, Mgst1, Gstt1, Gstt2, and Gstt3) for each sample.

In the same way, the PCA plots for maximal OCR and secreted cytokines from BMDM were made from the OXPHOS data of Seahorse analysis and ELISA data by calculating the percentage of cytokines

released from BMDM of lean and SerpinB2$^{+/+}$ mice with respect to obese and SerpinB2$^{-/-}$ mice.

## Chromatin immunoprecipitation

We performed ChIP as previously[169]. Briefly, SerpinB2 (ThermoFisher Scientific, # PA5-27857) and IgG antibodies were used for chromatin precipitation. Genomic DNA was extracted from immunoprecipitated and non-immunoprecipitated (INPUT) samples, and qPCR was performed to quantify the promoter binding regions.

## Statistics

The data were represented as mean ± SEM. The Mann Whitney test (two-tailed) was used to determine the significance between two groups. One-way ANOVA with Bonferoni's post hoc correction test was performed to determine differences among data obtained from more than two groups. Results were considered statistically significant when $P < 0.05$.

## Reporting summary

Further information on research design is available in the Nature Portfolio Reporting Summary linked to this article.

## Data availability

All data included in the Supplementary Information are available from the authors, as are unique reagents used in this article. The raw numbers for charts and graphs are available in the Source Data file whenever possible. The RNA sequencing data used in this study have been deposited in GEO under the accession code GSE118226. Source data are provided with this paper.

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

## Acknowledgements

This work was supported by National Institute of Health grants R00HL121076-03, R01HL14262, R01HL143967, R01AG069399, and R01DK129339, and AHA Transformational Project Award (19TPA34910142), AHA Innovative Project Awards (19IPLOI34760566 and 23IPA1053549) and ALA Innovation Project Award (IA-629694) (to PD). S.B.V. was supported by the AHA Postdoctoral Fellowship Award (20POST35210088) and the VMI postdoctoral scholar award. Confocal and intravital microscopy was performed using the NIH-supported microscopy resources in the Center for Biologic Imaging. Specifically, the confocal microscope was supported by the NIH grant 1S10OD019973-01. The graphical illustration and the cartoon in Fig. 3K were designed using BioRender software, and the mouse images were adapted from Servier Medical Art- Creative Commons Attribution 3.0 Unported License (https://creativecommons.org/licenses/by/3.0/).

## Author contributions

S.B.V. and S.S. were involved in designing and conducting experiments, data analysis, and writing the manuscript. M.A.U., J.S., E.J., A.M., J.F., L.L., A.M., and K.S.R. helped in conducting experiments. JS and MR provided us with the VAT samples from deceased donors. I.S., J.K., and M.J. performed hyperinsulinemic-euglycemic clamp studies. S.S. helped with Seahorse experiments. R.M.O. and V.Y. provided us with critical insights into the experiments involving mouse models of insulin resistance. P.D. was involved in designing research studies, conducting experiments, acquiring and analyzing data, and writing the manuscript.

## Competing interests

The authors declare no competing interests.

## Additional information

**Sathish Babu Vasamsetti**[1,2,11], **Samreen Sadaf**[1,11], **Mohammad A. Uddin**[1], **Jixing Shen**[1], **Ebin Johny** ®[1], **Awishi Mondal**[1], **Jonathan Florentin**[1], **Liqun Lei**[1], **Aleef Mannan**[1], **Krithika Sudhakar Rao**[1], **John Sembrat** ®[1], **Mauricio Rojas** ®[3], **Ian Sipula**[4], **Jake Kastroll**[4], **Michael J. Jurczak** ®[4,5], **Sruti Shiva**[1,6], **Robert M. O'Doherty**[1,4], **Vijay Yechoor** ®[1,4] & **Partha Dutta** ®[1,2,7,8,9,10] ✉

[1]Pittsburgh Heart, Lung, Blood, and Vascular Medicine Institute, University of Pittsburgh, Pittsburgh, PA, USA. [2]Division of Cardiology, Department of Medicine, University of Pittsburgh, Pittsburgh, PA, USA. [3]Division of Pulmonary, Critical Care, & Sleep Medicine, Davis Heart & Lung Research Institute, The Ohio State University, Columbus, OH, USA. [4]Division of Endocrinology and Metabolism, Department of Medicine, University of Pittsburgh, Pittsburgh, PA, USA. [5]Center for Metabolism and Mitochondrial Medicine, University of Pittsburgh, Pittsburgh, PA, USA. [6]Department of Pharmacology and Chemical

Biology, University of Pittsburgh, Pittsburgh, PA, USA. [7]Department of Immunology, University of Pittsburgh, Pittsburgh, PA, USA. [8]Department of Bioengineering, Swanson School of Engineering, University of Pittsburgh, Pittsburgh, PA, USA. [9]The Center for Cardiovascular Inflammation, University of Pittsburgh, Pittsburgh, PA, USA. [10]Veterans Affairs Pittsburgh Healthcare System, Pittsburgh, PA, USA. [11]These authors contributed equally: Sathish Babu Vasamsetti, Samreen Sadaf. ✉e-mail: duttapa@pitt.edu

