## [Transparent Peer Review file · Nature Communications]

Tissue-resident macrophage survival depends on mitochondrial function regulated by SerpinB2 in chronic inflammation

Corresponding Author: Dr Partha Dutta

Version 0:

Reviewer comments:

Reviewer #1

(Remarks to the Author)

The manuscript reads well and contains an impressive amount of data. It seems that the conclusions of the manuscript are partly supported by the results. I do have several questions and suggestions that may help to further improve the current manuscript.

1. I do wonder about the novelty of the first part of the manuscript related to resident versus infiltration macrophages in adipose tissue during the development of obesity. Most of these results part of the first figure of the manuscript are more or less established. I would suggest to shorten this part of the manuscript and focus on the role of serpinB2 in promoting the survival of tissue resident macrophages.
 2. I am somewhat confused about the source/role of IFN γ . It seems this originates from myeloid cells part of the adipose tissue, yet what cells would produce it? Newly recruited macrophages? Or could there also be a role for adipocytes or other non-immune resident cells? And what signals would drive upregulation of IFN γ production in adipose tissue?
 3. Why would the role of SerpinB2 be so specific for resident macrophages? Why would it not be expressed by non-resident macrophages? It also seems a well-orchestrated event, so why would this mechanism exist since it seems to negatively impact on adipose tissue health?
 4. How do these VAT resident macrophages restrain adipocyte growth? This is not clear from the data presented in the current manuscript.
 5. The authors should be careful in concluding on the role of resident vs. infiltrating macrophages and their role in systemic insulin sensitivity/resistance. This is all based on associations presented in figure 1.
 6. In figure 2, it would be important to indicate BW development of the animals after the depletion of macrophages. How does this procedure impact on macrophages in tissues other than the adipose tissue ?
 7. I do not really understand the added value of figure 3S. Yes, palmitate decreases SerpinB2 expression, yet do the authors now suggest that fatty acids are the drivers of decreased expression in adipose tissue resident macrophages?
 8. Could the authors provide more clinical characteristics of the individuals with obesity that were included in the manuscript?
 9. In figure 4H and 4I different metabolic and inflammatory genes are presented. Why are data presented in different layouts? And what is the added value of measuring these metabolic genes?
It would be helpful to learn more about the BW of the animals in the main figure and not in the supplemental figure 3.
 10. In figure 6 related to the metabolic profiles of the macrophages, it would be very informative to also present the ECAR data of these experiments. I would also suggest to show the complete OCR graphs in the main figure, so not only the maximal OCR. I am also somewhat confused about the data presented in figure 6, since all of these results are generated in ex vivo/in vitro experiments. This would imply that the absence of SerpinB2 would also promote apoptosis/changes in mitochondrial respiration in other tissue macrophages in vivo. Have the authors examined other tissues of their in vivo studies?
- In general, the manuscript contains an impressive amount of data. For certain figures, the authors might consider to lower data density helping readability of the manuscript. I would also suggest to remove the last part of the manuscript that is focused on adipocyte size. This seems of little added value in conveying the message that resident VAT adipose tissue macrophages control insulin sensitivity.
- The results section is very extensive, yet the discussion is short and more or less a summary of the results. I think it also reflects the different results that are presented in the current manuscript. The third paragraph shortly deals with how

macrophages control adipocyte growth, while the next paragraph discuss the role of mitochondria in cytokine production by macrophages without a clear connection between both paragraphs.

Reviewer #2

(Remarks to the Author)

The current study proposed to understand the impact of SerpinB2 on tissue-resident macrophage survival in obese adipose tissue. A set of transgenic mouse models were employed to determine the importance of SerpinB2 in CX3CR1+CCR2+ or CX3CR1+CCR2- macrophages. The authors first examined the two subsets of macrophages in visceral adipose tissue (VAT) transcriptome profiles and subsequently focused on SerpinB2-regulated macrophage survival and their impact on systemic metabolic regulation. The study also zoomed in on SerpinB2-regulated oxidative stress mediated by mitochondria releasing cytochrome C as the molecular mechanism. The overall study is interesting and likely to provide new aspects of how SerpinB2 regulates resident macrophage function in obese visceral adipose tissue. Although some aspects of the topic were supported by detailed characterization, major concerns should be addressed to support the primary argument as follows:

1. One key aspect of the study is to argue that SerpinB2 will selectively regulate residential vs monocyte-derived macrophage functions in obese VAT. The study focused on these two macrophage subsets marked with CX3CR1 and CCR2 and suggested that CX3CR1+CCR2+ are monocyte derived macrophages while CX3CR1-CCR2- are residential macrophages. However, this key point as shown in Fig 1 was rather not convincing. For example, the expression pattern of CCR2 in CX3CR1-GFP positive vs negative portions are comparative evidenced by MFI. As shown in Fig S1F-- a significant portion of the CX3CR1-CCR2+ population was not calculated in both lean and obese samples. These cells appear to have similar CCR2 MFI as those in CX3CR1-GFP+ populations.
2. The comparison between CX3CR1+CCR2+/- CX3CR1-CCR2- to CD206+/CD206+ cells further argue the indicated populations could be more suitable for inflammatory vs anti-inflammatory states than residential or monocyte-derived macrophages. Furthermore, a good body of evidence suggest that monocyte-derived macrophage can be either inflammatory or anti-inflammatory.
3. Authors suggested that a "decrease" in CCR2- population is likely due to increased apoptosis. However, almost all results on this point were presented as proportion differences, which could be merely due to a significant increase in the CCR2+ population. Meanwhile, authors only reported some minor increase in the proportion of caspase 3+ cells in the CCR2- population, but no results were provided for CCR2+ cells. Without considering obesity-induced stress in this context, data presented only on the CCR2 population are not convincing. Similarly, no results on the CCR2+ population were examined. In Fig 2B, no differences were observed in CCR2+ cells, and the authors proposed that it was likely due to high turnovers. With a significantly increased infiltration rate, it could also accompanied by a high rate of apoptosis.
4. SerpinB2 is expressed in monocytes and macrophages and can be further induced under inflammatory conditions, as reported previously. Results in this study also reported that THP1 differentiated mac in vitro can express high levels of SerpinB2. Why was its expression in CCR2+ mac barely detectable (Fig 3)?
5. SerpinB2-deficient macrophages are apoptotic, which could have various causal factors, including a direct impact on the inflammasome. The rationale for only examining cytochrome C levels was not clear.

Reviewer #3

(Remarks to the Author)

The manuscript by Vasamsetti et al explores the dynamics of adipose tissue macrophages (ATMs) during obesity and highlights the role of SerpinB2 in obesity related macrophage dysfunction. Using a mouse model of diet induced obesity they provide evidence that during obesity "resident" ATMs undergo apoptosis. Using an inducible model of macrophage cell death they suggest that loss of resident macrophages worsening obesity and insulin resistance. One of the genes enriched in "resident" ATMs was SerpinB2. The expression of SerpinB2 was suppressed by IFN γ in culture and KO of the IFN γ R in myeloid cells increased in the number of SerpinB2 positive macrophages and improved measures of insulin resistance in vivo. KO of SerpinB2 also led to increased ATM OXPHOS, adipose inflammation and worse insulin resistance, which could be rescued by IL-4. The authors provide evidence that Serpin is associated with improved mitochondrial function and less ROS generation. Overall, the study tackles an important area of research. In addition, the authors should be commended for their combined use of several genetic models and inhibitors to address the role of SerpinB2 in ATMs. However, there are several points that should be addressed to improve the impact of the study.

Major Comments:

- 1) The authors needs be careful with the terminology of resident vs recruited adipose macrophages purely based on expression of CX3CR1 and/or CCR2. Although CX3CR1 and CCR2 are expressed on incoming recruited cells they can be downregulated rapidly in the tissue. Therefore, cells that are CX3CR1 and CCR2 negative can still be monocyte-derived, recruited cells. The "true" ATM resident population is very small in number (~ 10,000 cells per fat pad) whereas upon high fat feeding the number of ATMs increases 10-fold and typically consists of CD9, TREM2 expressing ATMs (known as lipid

associated macrophages (LAMS; PMID31257031), which are monocyte derived. To address the conversion rate of Cx3Cr1 pos to Cx3Cr1 neg macrophages in adipose tissue during obesity, the authors should use their Cx3Cr1-YFP, Cx3CR1-CreER-TdT system for fate mapping. Using these mice, the authors could determine what percentage of the "red" cells (MdMs) are YFP-positive (i.e. actively expressing Cx3Cr1) at day 1 and day 7 post labeling with tamoxifen. In addition, it would be useful to assess for CD9 expression via flow cytometry to identify "lipid associated macrophages" and TIM4 to assess for true resident macrophages. The authors also need to include the absolute number of cells per gram of tissue or per fat pad in addition to just percentages. This information is critical for interpreting the study and understanding adipose tissue macrophage dynamics.

2) In Figure 2, the authors employ an innovative approach to delete macrophages from the adipose tissue of obese mice by leveraging a Cre-activatable expression of a photosensitive channel that triggers cells death upon exposure to blue light. However, the experimental design indicates that mice were fed a HFD for 1 month prior to exposure to light. During the month of feed the ATM compartment expands substantially via recruitment and proliferation of monocyte derived cells. As discussed in point #1, the majority of the macrophages in the adipose tissue at this time are likely CD9-positive LAMs (which are predominantly CCR2 low). As LysM expresses well in all macrophages but only poorly in monocytes (< 50%) it is possible that the failure of light to deplete CCR2 cells is both related to rapid turnover and poor expression in monocytes/early macrophages. Either way the use of CCR2 IF to assess the specificity the deletion approach for resident vs. recruited cells is flawed. Flow cytometry using other markers such as CD9 and TIM4 should be employed. My suspicion is that this approach leads to the loss of LAMs, which are recruited macrophages that play a protective role in obese adipose tissue and liver (PMID 31257031; PMID36521495) . Without this information it is hard to harmonize this study with the larger literature around ATMs and obesity.

3) In general, the figures are difficult to follow even after reading the legend. As an example, in Figure 1 the A panel is hard to interpret as the Cx3CR1 pos dot plot contains 80% Cx3Cr1 negative cells. Panels B and C which included the tamoxifen labeling system and parabiosis require going back to the main text to figure out what these plots represent. Adding schematics and/or better details in the legend is important. In addition, removing some of the panels from Figure 1 would improve the impact of the data. For example, the value of Panel D and Panel F are not clear. This data could be in supplemental. Also, for panels M and N the protocol for when tamoxifen was given relative to the start of HFD is unclear. How many does were given? Tamoxifen can also influence metabolic phenotypes so this needs to be accounted for.

4) What happened to total ATM content in the myeloid-IFNGR1 KO mice? Were total numbers of macrophages impacted or just the macrophage phenotype? Using flow cytometry to assess SerpinB2 expression in WT vs. KO ATMs would enhance strength the semiquantitative IF data in Fig. 4 C (the provide flow data for SerpinB2 later in the manuscript showing feasibility).

5) The NAC experiments are somewhat hard to interpret as this is a systemic antioxidant. What happened to macrophage number and cell death with NAC? NAC is also poorly tolerated as a oral solution due to its smell. Could some of the observed effects be related to changes in weight or diet intake?

6) The authors use a myeloid COX10 KO mouse to provide evidence that enhanced OXPHOS may be detrimental to ATMs and lead to increased ROS and death. However, this data is only shown in supplemental and is missing the seahorse data to show the impact of COX10 KO on BMDM metabolism.

Minor:

1) In Figure 1 N why did the authors use autofluorescence? Macrophage specific staining with either CD68 or F4/80 would be better here.

2) As mentioned in major comments panel 1M should include both percentage and total number of macrophages as this will be hugely different between STD vs. HFD fed animals.

3) The authors should include the flow gating scheme with representative images for ATMs in the supplemental data.

4) In figure 5 K they authors show Mitosox staining, which reveals subtle differences between WT and SerpinB2 KO macrophages. They then report a ratio of Mitosox to Mitotracker. Was this based on flow determined MFI values? Does this mean that mitochondrial number was also different between genotypes to drive the ratio up?

5) The protocol for IL-4 reconstitution is missing from the methods. Did they use IL-4/IL-4R complexes or just naked cytokine?

6) In panel 5M the displayed MFI quantification is not congruent with the adjacent flow plot. The numbers for MFI must x 1000. This should be indicated.

Version 1:

Reviewer comments:

Reviewer #1

(Remarks to the Author)

The authors have adequately responded to many of the suggestions. However, I do still have several remaining issues. The manuscript still remains (too) information dense making it hard to read and digest all data that is presented. This also seems to go hand in hand with the fact that the manuscript basically presents two stories as one. The part on adipocyte size regulation seems a somewhat different story and is also not mentioned in the abstract. I would advise to reconsider this part again.

I also feel that the inclusion of the data obtained from human VAT samples is somewhat difficult to interpret. These samples were obtained from deceased people introducing many potential confounders. Also, age and BMI is hugely variable and no information on cause of death is presented.

Note, there is also no need to present BMI with 3 decimals. Also, in the M&M section the study participants are described as people with and without diabetes, while in the results section the participants are presented as lean vs. obese. Some of the study participants do have diabetes, yet based on their BMI would be classified as overweight. Also, how is diabetes being classified and is this type 1 or type 2 diabetes?

Related to the animal study with the LysM $+/+$ *Ifn γ 1* fl/fl and LysM $cre^{-/+}$ *Ifn γ 1* fl/fl mice, some results are difficult to interpret. Although total BW is similar, VAT adipose tissue is smaller. What about total adipose tissue mass? Anything known about activity, food intake of the animals? I guess multiple factors are at play here beyond the local effects of the macrophages in VAT.

Reviewer #2

(Remarks to the Author)

The authors have provided additional results and discussion, addressing each point raised in the first review, particularly the attributes used to define "residential" versus "infiltration" macrophages. I have no further questions for the current version.

Reviewer #3

(Remarks to the Author)

In the revised manuscript the authors have largely addressed my concerns. The added significant new data and reorganized the manuscript, which significantly improves the impact of the study. I have no further comments.

REVIEWER COMMENTS

Reviewer #1 (Remarks to the Author):

The manuscript reads well and contains an impressive amount of data. It seems that the conclusions of the manuscript are partly supported by the results. I do have several questions and suggestions that may help to further improve the current manuscript.

We appreciate the overall enthusiasm of this reviewer for our study. We have addressed each comment of this reviewer with additional data and explanation as discussed below.

1. I do wonder about the novelty of the first part of the manuscript related to resident versus infiltration macrophages in adipose tissue during the development of obesity. Most of these results part of the first figure of the manuscript are more or less established. I would suggest to shorten this part of the manuscript and focus on the role of serpinB2 in promoting the survival of tissue resident macrophages.

As this reviewer recommended, we have now shortened the part of the manuscript containing the functions of resident and monocyte-derived macrophages to focus on the role of SerpinB2 in promoting the survival of tissue resident macrophages (lines 139-164).

2. I am somewhat confused about the source/role of IFN γ . It seems this originates from myeloid cells part of the adipose tissue, yet what cells would produce it? Newly recruited macrophages? Or could there also be a role for adipocytes or other non-immune resident cells? And what signals would drive upregulation of IFN γ production in adipose tissue?

To find the source of IFN- γ , we stained VAT of lean and obese mice with the markers for adipocytes, fibroblasts, and macrophages. We observed that obesity significantly increased the frequencies of IFN- γ -expressing fibroblasts and macrophages (Response Fig. 1 and Fig. 4C and lines 282-283 in the manuscript).

Several mechanisms have been proposed for the production of IFN- γ by obese adipose tissue. In obesity, adipocytes secrete CXCL12, which recruits natural killer cells and stimulates the secretion of IFN- γ by these cells¹. Besides natural killer cells, inflammatory macrophages and T helper cells can infiltrate adipose tissue and secrete IFN- γ in obesity^{2,3}. Additionally, the activation of HIF-1 α

in response to the hypoxic environment in adipose tissue in obesity can stimulate the transcription of various genes including *Ifng*⁴. We have now included these mechanisms of IFN- γ production in obesity in Discussion (lines 511-516 in the manuscript).

3. Why would the role of SerpinB2 be so specific for resident macrophages? Why would it not be expressed by non-resident macrophages? It also seems a well-orchestrated event, so why would this mechanism exist since it seems to negatively impact on adipose tissue health?

Our data provided in the manuscript demonstrate that IFN- γ suppresses SerpinB2 expression in macrophages (Fig. 4). Based this observation and the data showing that monocyte-derived non-resident macrophages express low levels of SerpinB2, we hypothesized that this macrophage population has high levels of the IFN- γ receptors. Indeed, this population has high expression of *Ifngr1* and *Ifngr2* compared to VAT resident macrophages (Response Fig. 2 and Fig. 4O, and lines 304-305 in the manuscript).

The question of this reviewer regarding the significance of SerpinB2 expression by the macrophage subsets is interesting. We have observed that IFN- γ , which is released in high quantity in infection, decreases SerpinB2 levels in VAT resident macrophages, resulting in their apoptosis. Loss of VAT resident macrophages can lead to infiltration of monocyte-derived macrophages, which have better infection fighting ability and lower SerpinB2 expression, thereby are resistant to IFN- γ -mediated apoptosis. Future studies will be required to examine the significance of SerpinB2 expression by the macrophage subsets in infection clearance. We have now included this point in Discussion (lines 516-522 in the manuscript).

4. How do these VAT resident macrophages restrain adipocyte growth? This is not clear from the data presented in the current manuscript.

We appreciate this insightful comment of this reviewer. Adipocyte expansion is regulated by several biological processes including adipocyte apoptosis and necrosis, adipogenesis, lipolysis, and lipogenesis. Inflammatory cytokines produced by macrophages and monocytes can regulate these processes. A recent study showed that PDDGcc produced by macrophages can control adipocyte energy storage, which aids in the growth of adipose tissue in obesity⁵. We observed that adipose monocyte-derived and tissue-resident macrophages expressed differential amounts of cytokines affecting adipose growth. For example, adipose resident macrophages express low levels of *Il1b*, which stimulates adipocyte growth⁶, and high levels of

Tgfb2, which inhibits adipocyte growth⁷, compared to monocyte-derived macrophages (Response Fig. 3 and Fig. S7B and lines 444-446 in the manuscript). However, how these inflammatory cues regulate adipocyte growth by affecting adipocyte apoptosis and necrosis, adipogenesis, lipolysis, or lipogenesis is beyond the scope of this study. We have now acknowledged this limitation of our study in Discussion (lines 558-566 in the manuscript).

5. The authors should be careful in concluding on the role of resident vs. infiltrating macrophages and their role in systemic insulin sensitivity/resistance. This is all based on associations presented in figure 1.

This point of the reviewer is well taken. We have specifically depleted VAT resident macrophages in an optogenetic *LysM^{cre/+} Chr2^{fl/fl}* mice and found that the depletion of this population led to systemic glucose intolerance and insulin resistance (Fig. 2).

Additionally, we have acknowledged the correlative nature of the data presented in Figure 1 in Discussion (lines 569-572 in the manuscript).

6. In figure 2, it would be important to indicate BW development of the animals after the depletion of macrophages. How does this procedure impact on macrophages in tissues other than the adipose tissue?

We now provide the body weight data in these mice (Figure S2E). Additionally, we have examined if blue light exposure in VAT of the optogenetic mice alters the expression of Annexin V, an early marker of apoptosis, in macrophages residing in distant organs. This treatment did not significantly change Annexin V expression in microglia, Kupffer cells, and splenic macrophages (Response Fig. 4A and Fig. S2C and lines 226-229 in the manuscript). Consistently, we observed no significant difference in total hepatic macrophage and Kupffer cell numbers (Response Fig. 4B and Fig. S2D and lines 229-230 in the manuscript).

7. I do not really understand the added value of figure 3S. Yes, palmitate decreases SerpinB2 expression, yet do the authors now suggest that fatty acids are the drivers of decreased expression in adipose tissue resident macrophages?

We have used palmitate as a stimulus for macrophages. It has been reported that obesity can increase endogenous palmitic acid levels⁸. This excess palmitic acid can promote inflammation⁹. Because exaggerated inflammation can reduce SerpinB2 expression, we sought to test whether

palmitic acid can reduce the expression of this gene. We have now clarified this rationale in the text (lines 272-275 in the manuscript).

8. Could the authors provide more clinical characteristics of the individuals with obesity that were included in the manuscript?

Since these patients were deidentified deceased donors, we have limited demographic and clinical information for these patients. However, we were able to retrieve the information on age, sex, BMI, and diabetes status in these patients (Response Table 1 and Table S1 and lines 267 in the manuscript).

9. In figure 4H and 4I different metabolic and inflammatory genes are presented. Why are data presented in different layouts? And what is the added value of measuring these metabolic genes? It would be helpful to learn more about the BW of the animals in the main figure and not in the supplemental figure 3.

We appreciate this comment of the reviewer. The metabolic genes shown in new Fig. 4I are insulin-sensitivity promoting genes, which were increased in the absence of *Ifngr1* in myeloid cells. New Fig. 4J shows the inflammatory genes, which were suppressed in the absence of myeloid *Ifngr1*. As this reviewer suggested, we have now moved the body weight data to the main figure (Fig. 4F).

10. In figure 6 related to the metabolic profiles of the macrophages, it would be very informative to also present the ECAR data of these experiments. I would also suggest to show the complete OCR graphs in the main figure, so not only the maximal OCR. I am also somewhat confused about the data presented in figure 6, since all these results are generated in ex vivo/in vitro experiments. This would imply that the absence of SerpinB2 would also promote apoptosis/changes in mitochondrial respiration in other tissue macrophages in vivo. Have the authors examined other tissues of their in vivo studies?

Patient	Age	Sex	BMI	Diabetic
HAC-2	49	F	29.737	No
HAC-4	51	M	27.31	No
HAC-6	91	M	28.1	Yes
HAC-10	22	F	20.906	No
HAC-12	25	F	25.712	Yes
HAC-13	59	F	33.299	Yes
HAC-15	56	F	28.125	No
HAC-17	19	M	22.531	No
HAC-18	65	M	29.037	No
HAC-19	84	F	22.319	No
HAC-20	73	M	34.6	Yes
HAC-22	42	F	45.54	Yes
HAC-26	59	M	28	No
HAC-28	19	F	20.5	No
HAC-30	83	M	22.4	No
HAC-31	85	F	20.05	No
HAC-32	65	F	41.025	Yes
HAC-33	76	M	25.6	Yes
HAC-38	57	M	39	No

Response Table 1: The demographic information of patients. Mesenteric adipose tissues from deceased donors were collected. The demographic information of these patients is provided in this table.

We have tried to measure OCR and ECAR in macrophages sorted from various organs. However, we were not successful in measuring these parameters from these sorted macrophages because they are metabolically inactive just after the sort probably due to the stress of cell sorting. We have now provided the OCR data in the main figure and ECAR traces in the supplement (Response Fig. 5 and Fig. 6C and S5F and line 372 in the manuscript). Additionally, our new data demonstrated that Kupffer cell numbers in the liver do not change in *SerpinaB2*^{-/-} obese mice (Response Fig. 6 and Fig. S4B and lines 312-313 in the manuscript). The mechanisms of the absence of numerical changes in Kupffer cells of these mice should be addressed by future studies (Discussion, lines 575-577 in the manuscript).

11. In general, the manuscript contains an impressive amount of data. For certain figures, the authors might consider to lower data density helping readability of the manuscript. I would also suggest to remove the last part of the manuscript that is focused on adipocyte size. This seems of little added value in conveying the message that resident VAT adipose tissue macrophages control insulin sensitivity. As this reviewer recommended, we have now extensively worked on the manuscript and figures to make them less dense. However, since the editor wanted us to keep the adipocyte data, we have moved the data involving adipocyte characteristics after NAC treatment to the supplement (Fig. S7K-M) to focus on the message that resident VAT adipose tissue macrophages control insulin sensitivity.

12. The results section is very extensive, yet the discussion is short and more or less a summary of the results. I think it also reflects the different results that are presented in the current manuscript. The third paragraph shortly deals with how macrophages control adipocyte growth, while the next paragraph discusses the role of mitochondria in cytokine production by macrophages without a clear connection between both paragraphs.

We appreciate this reviewer for this excellent comment. We have now worked on the discussion extensively to clearly connect different parts of this section.

Reviewer #2 (Remarks to the Author):

The current study proposed to understand the impact of SerpinB2 on tissue-resident macrophage survival in obese adipose tissue. A set of transgenic mouse models were employed to determine the importance of SerpinB2 in CX3CR1+CCR2+ or CX3CR1-CCR2- macrophages. The authors first examined the two subsets of macrophages in visceral adipose tissue (VAT) transcriptome profiles and subsequently focused on SerpinB2-regulated macrophage survival and their impact on systemic metabolic regulation. The study also zoomed in on SerpinB2-regulated oxidative stress mediated by mitochondria releasing cytochrome C as the molecular mechanism. The overall study is interesting and likely to provide new aspects of how SerpinB2 regulates resident macrophage function in obese visceral adipose tissue. Although some aspects of the topic were supported by detailed characterization, major concerns should be addressed to support the primary argument as follows:

We are glad to know that this reviewer found this study interesting. We have addressed the crucial concerns of this reviewer with new experimental data.

1. One key aspect of the study is to argue that SerpinB2 will selectively regulate residential vs monocyte-derived macrophage functions in obese VAT. The study focused on these two macrophage subsets marked with CX3CR1 and CCR2 and suggested that CX3CR1+CCR2+ are monocyte derived macrophages while CX3CR1-CCR2- are residential macrophages. However, this key point as shown in Fig 1 was rather not convincing. For example, the expression pattern of CCR2 in CX3CR1-GFP positive vs negative portions are comparative evidenced by MFI. As shown in Fig S1F-- a significant portion of the CX3CR1-CCR2+ population was not calculated in both lean and obese samples. These cells appear to have similar CCR2 MFI as those in CX3CR1-GFP+ populations.

The comment of this reviewer regarding CCR2 expression in CX₃CR1⁺ cells is well taken. To distinguish the macrophage subsets in mouse VAT based on their CCR2 and CX₃CR1 expression, we have now shown the gates for the following populations in Fig. S1H: CX₃CR1⁺ CCR2⁺, CX₃CR1⁻ CCR2⁻, and CX₃CR1⁻ CCR2⁺. We have also calculated the frequencies and numbers of CX₃CR1⁻ CCR2⁺ macrophages in lean and obese mice. Although the frequencies of these macrophages decreased, their numbers were not significantly changed in obese mice (Response

Fig. 7 and Fig. S1J and lines 173-175 in the manuscript). We focused on CX₃CR1⁺ CCR2⁺ and CX₃CR1⁻ CCR2⁻ VAT macrophages since the abundance of these macrophage subsets was elevated and subsided in obesity, respectively (Response Fig. 8A and Fig. S1I and line 173 in the manuscript).

2. The comparison between CX₃CR1⁺CCR2⁺/ CX₃CR1⁻CCR2⁻ to CD206⁺/CD206⁻ cells further argue the indicated populations could be more suitable for inflammatory vs anti-inflammatory states than residential or monocyte-derived macrophages. Furthermore, a good body of evidence suggest that monocyte-derived macrophage can be either inflammatory or anti-inflammatory.

We agree with this interpretation that CX₃CR1⁺ CCR2⁺ and CX₃CR1⁻ CCR2⁻ macrophages are inflammatory and anti-inflammatory, respectively. We have now also included this information in Results (lines 161-163 in the manuscript). Additionally, using a mouse model of parabiosis, we demonstrate that CX₃CR1⁻ CCR2⁻ macrophages are adipose residents whereas CX₃CR1⁺ CCR2⁺ macrophages are hematopoietic cell-derived (Fig. 1C and line 141 in the manuscript). Furthermore, we previously used an array of techniques, including cell lineage tracing, to ascertain that CX₃CR1⁻ CCR2⁻ macrophages are adipose residents whereas CX₃CR1⁺ CCR2⁺ macrophages are monocyte-derived¹⁰.

3. Authors suggested that a “decrease” in CCR2⁺ population is likely due to increased apoptosis. However, almost all results on this point were

presented as proportion differences, which could be merely due to a significant increase in the CCR2⁺ population. Meanwhile, authors only reported some minor increase in the

proportion of caspase 3+ cells in the CCR2- population, but no results were provided for CCR2+ cells. Without considering obesity-induced stress in this context, data presented only on the CCR2 population are not convincing. Similarly, no results on the CCR2+ population were examined. In Fig 2B, no differences were observed in CCR2+ cells, and the authors proposed that it was likely due to high turnovers. With a significantly increased infiltration rate, it could also be accompanied by a high rate of apoptosis.

As this reviewer suggested, we have now included the absolute numbers of various macrophage subsets in VAT of lean and obese mice (Response Fig. 8A and Fig. S1I and line 173 in the manuscript). We have also shown the dynamics of the numbers of other adipose macrophage subsets reported in the literature, including CD9+ lipid-associated macrophages and TIM4+ resident macrophages (Response Fig. 8B & 8C and Fig. S1K and lines 175-180 in the manuscript). The number of both subsets of macrophages increased significantly at one month after HFD. However, although CD9-expressing lipid-associated macrophages remained significantly increased at two months after HFD, the abundance of TIM4+ VAT resident macrophages diminished at this time. Our data indicate that, although there is an initial expansion of the TIM4+ resident macrophage subset after HFD initiation, this macrophage population shrinks at latter stages.

Additionally, we have shown that the frequency of caspase 3+ CCR2+ macrophages did not significantly change in obese mice by flow cytometry (Response Fig. 9A and Fig. 1Q and lines 191-192 in the manuscript).

Moreover, as this reviewer predicted, blue light exposure in obese *LysM^{cre/+} ChR2^{fl/fl}* mice significantly enhanced Annexin V expression, a marker of early apoptosis, in CX₃CR1+ CCR2+ VAT macrophages as determined by flow cytometry (Response Fig. 9B and Fig. S2A and lines 219-221 in the manuscript). We have now included this information in the manuscript (lines 217-218).

4. SerpinB2 is expressed in monocytes and macrophages and can be further induced under inflammatory conditions, as reported previously. Results in this study also reported that THP1 differentiated mac in vitro can express high levels of SerpinB2. Why was its expression in CCR2+ mac barely detectable (Fig 3)?

Although we observed high expression of SerpinB2 by VAT resident macrophages, CCR2+ macrophages also expressed SerpinB2, albeit at low levels. We found that this subset of macrophages expresses high levels of the IFN- γ receptors (Response Fig. 2 and Fig. 4O and lines 304-305 in the manuscript). Since we have shown that IFN- γ reduces SerpinB2 expression in macrophages (Fig. 4N), low SerpinB2 expression by CCR2+ macrophages may be due to their high expression of the IFN- γ receptors. Additionally, the discrepancies of SerpinB2 expression by

THP1-derived macrophages and VAT monocyte-derived macrophages could be attributed to altered phenotypes of cultured macrophages. In line with this hypothesis, previous studies have shown that macrophages cultured *in vitro* exhibit significant changes in their epigenome^{11, 12, 13}.

5. SerpinB2-deficient macrophages are apoptotic, which could have various causal factors, including a direct impact on the inflammasome. The rationale for only examining cytochrome C levels was not clear.

We focused on cytochrome C because we observed SerpinB2 modulates mitochondrial function, which can regulate cytochrome C release from mitochondria. Additionally, we observed no change in the expression of Nlrp3, which is involved in inflammasome signaling, in *SerpinB2*^{-/-} adipose resident macrophages compared to the control (Response Fig. 10 and Fig. S4F and lines 352-356 in the manuscript). We have now discussed this in Results and acknowledged the limitations of not studying inflammasomes in *SerpinB2*-deficient macrophage apoptosis in Discussion (lines 498-500 in the manuscript).

Reviewer #3 (Remarks to the Author):

The manuscript by Vasamsetti et al explores the dynamics of adipose tissue macrophages (ATMs) during obesity and highlights the role of SerpinB2 in obesity related macrophage dysfunction. Using a mouse model of diet induced obesity they provide evidence that during obesity “resident” ATMs undergo apoptosis. Using an inducible model of macrophage cell death they suggest that loss of resident macrophages worsening obesity and insulin resistance. One of the genes enriched in “resident” ATMs was SerpinB2. The expression of SerpinB2 was suppressed by IFN γ in culture and KO of the IFN γ R in myeloid cells increased in the number of SerpinB2 positive macrophages and improved measures of insulin resistance *in vivo*. KO of SerpinB2 also led to increased ATM OXPHOS, adipose inflammation and worse insulin resistance, which could be rescued by IL-4. The authors provide evidence that Serpin is associated with improved mitochondrial function and less ROS generation. Overall, the study tackles an important area of research. In addition, the authors should be commended for their combined use of several genetic models and inhibitors to address the role of SerpinB2 in ATMs. However, there are several points that should be addressed to improve the impact of the study.

We appreciate the positive evaluation of our study by this reviewer. We have extensively revised the manuscript with additional data and explanation from the literature to address the major and minor concerns mentioned below.

Major Comments:

1) The authors need to be careful with the terminology of resident vs recruited adipose macrophages purely based on expression of CX3CR1 and/or CCR2. Although CX3CR1 and CCR2 are expressed on incoming recruited cells they can be downregulated rapidly in the tissue. Therefore, cells that are CX3CR1 and CCR2 negative can still be monocyte-derived, recruited cells. The “true” ATM resident population is very small in number (~ 10,000 cells per fat pad) whereas upon high fat feeding the number of ATMs increases 10-fold and typically consists of CD9, TREM2 expressing ATMs (known as lipid associated macrophages (LAMs; PMID31257031), which are monocyte derived. To address the conversion rate of CX3CR1^{pos} to CX3CR1^{neg} macrophages in adipose tissue during obesity, the authors should use their CX3CR1-YFP, CX3CR1-CreER-TdT system for fate mapping. Using these mice, the authors could determine what percentage of the “red” cells (Mdm) are YFP-positive (i.e. actively expressing CX3CR1) at day 1 and day 7 post labeling with tamoxifen. In addition, it would be useful to assess for CD9 expression via flow cytometry to identify “lipid associated macrophages” and TIM4 to assess for true resident macrophages. The authors also need to include the absolute number of cells per gram of tissue or per fat pad in addition to just percentages. This information is critical for interpreting the study and understanding adipose tissue macrophage dynamics.

Response Fig. 11: Most monocyte-derived macrophages do not downregulate CX₃CR1 in adipose tissue. CX₃CR1^{CreER/+} ROSA^{tdTomato/+} mice were injected with tamoxifen, and the flow cytometry analysis was performed four weeks later. n=3/ group.

We appreciate the comment of this reviewer. We performed a parabiosis (Fig. 1C) experiment to examine the contribution of monocytes to the differentiation of these two subsets of adipose macrophages. Additionally, as this reviewer suggested, we conducted a lineage tracing experiment in CX₃CR1-CreER-TdTomato mice (Response Fig. 11 and Fig. S1B and lines 143-147 in the manuscript). These mice were injected with tamoxifen. Four weeks after tamoxifen injection, most tdTomato⁺ macrophages were CX₃CR1^{YFP+}. Only about 12% of them were CX₃CR1^{YFP-}. These data indicate that CX₃CR1⁺ macrophages convert to CX₃CR1⁻ macrophages, albeit at low levels.

As this reviewer suggested, we have now shown the absolute numbers of the VAT macrophage subsets per mg of tissue (Response Fig. 8A and Fig. S1I and lines 171-173 in the manuscript). The number of CX₃CR1⁺ CCR2⁺ (monocyte-derived) macrophages increased while the abundance of CX₃CR1⁻ CCR2⁻ (resident) VAT macrophages was diminished in obese mice. We also assessed CD9-expressing lipid-associated macrophages and TIM4⁺ VAT resident macrophages at one and two months after HFD initiation. The number of both subsets of macrophages increased significantly at one month after HFD (Response Fig. 8B & 8C and Fig. S1K and lines 175-180 in the manuscript). However, although CD9-expressing lipid-associated macrophages remained significantly increased at two months after HFD, the abundance of TIM4⁺ VAT resident

macrophages diminished at this time. Our data indicate that although there is an initial expansion of the TIM4⁺ resident macrophage subset after HFD initiation, this macrophage population shrinks at latter stages possibly due to their high apoptosis. The flow cytometry plots (Response Fig. 8B) have not been included in the manuscript due to lack of space.

2) In Figure 2, the authors employ an innovative approach to delete macrophages from the adipose tissue of obese mice by leveraging a Cre-activatable expression of a photosensitive channel that triggers cells death upon exposure to blue light. However, the experimental design indicates that mice were fed a HFD for 1 month prior to exposure to light. During the month of feed the ATM compartment expands substantially via recruitment and proliferation of monocyte derived cells. As discussed in point #1, the majority of the macrophages in the adipose tissue at this time are likely CD9-positive LAMs (which are predominantly CCR2 low). As LysM expresses well in all macrophages but only poorly in monocytes (< 50%) it is possible that the failure of light to deplete CCR2 cells is both related to rapid turnover and poor expression in monocytes/early macrophages. Either way the use of CCR2 IF to assess the specificity the deletion approach for resident vs. recruited cells is flawed. Flow cytometry using other markers such as CD9 and TIM4 should be employed. My suspicion is that this approach leads to the loss of LAMs, which are recruited macrophages that play a protective role in obese adipose tissue and liver (PMID 31257031; PMID36521495). Without this information it is hard to harmonize this study with the larger literature around ATMs and obesity.

To address this crucial concern, we have performed confocal microscopy on VAT samples from these mice. Our new data demonstrate that TIM4⁺ macrophage abundance was lower in VAT of *LysM^{cre/+} ChR2^{fl/fl}* mice after blue light exposure as expected (Response Fig. 12 and Fig. S2B and lines 221-226 in the manuscript). Moreover, as this reviewer suspected, the number of CD9⁺ LAMs diminished significantly upon blue light exposure. We have now included this information in Results^{14, 15} (Lines 221-226 in the manuscript).

3) In general, the figures are difficult to follow even after reading the legend. As an example, in Figure 1 the A panel is hard to interpret as the Cx3CR1 pos dot plot contains 80% Cx3Cr1 negative cells. Panels B and C which included the tamoxifen labeling system and parabiosis require going back to the main text to figure out what these plots represent. Adding

schematics and/or better details in the legend is important. In addition, removing some of the panels from Figure 1 would improve the impact of the data. For example, the value of Panel D and Panel F are not clear. This data could be in supplemental. Also, for panels M and N the protocol for when tamoxifen was given relative to the start of HFD is unclear. How many does were given? Tamoxifen can also influence metabolic phenotypes so this needs to be accounted for.

We apologize for this confusion. Fig. 1A shows CX₃CR1-GFP⁺ macrophages in CX₃CR1^{GFP/+} mice that express GFP under the CX₃CR1 promoter. Panels B and C show the data from the cell lineage tracing and parabiosis experiments, respectively. As this reviewer suggested, we have now included experimental details in the figure legends. We have included the tamoxifen dosing regime in the figure legend. Finally, to address Comment #1 of Reviewer 1 on Figure 1, we have now shortened the part of the manuscript containing the functions of resident and monocyte-derived macrophages to focus on the role of SerpinB2 in promoting the survival of tissue resident macrophages (lines 140-164 in the manuscript).

4) What happened to total ATM content in the myeloid-IFNGR1 KO mice? Were total numbers of macrophages impacted or just the macrophage phenotype? Using flow cytometry to assess SerpinB2 expression in WT vs. KO ATMs would enhance strength the semiquantitative IF data in Fig. 4 C (the provide flow data for SerpinB2 later in the manuscript showing feasibility).

Our data indicate that the total number of adipose macrophages did not alter in obese *LysM^{+/+} Ifngr1^{fl/fl}* and *LysM^{Cre/+} Ifngr1^{fl/fl}* mice (Response Fig. 13 and Fig. S4A and lines 288-289 in the manuscript).

Because we were given a short period to revise this manuscript, we were unable to re-breed *LysM^{+/+} Ifngr1^{fl/fl}* and *LysM^{Cre/+} Ifngr1^{fl/fl}* mice to perform the flow cytometry experiment assessing SerpinB2 expression as suggested by the reviewer. However, we have shown a blinded analysis of the IF data, which demonstrates that SerpinB2 expression was significantly upregulated in the obese mice lacking myeloid *Ifngr1* (Figure 4D).

5) The NAC experiments are somewhat hard to interpret as this is a systemic antioxidant. What happened to macrophage number and cell death with NAC? NAC is also poorly tolerated as an oral solution due to its smell. Could some of the observed effects be related to changes in weight or diet intake?

As this reviewer requested, we have provided total and apoptotic macrophage numbers in mice after NAC treatment. We observed that, although NAC treatment did not alter the total number of

adipose macrophages, it reduced the abundance of macrophages expressing caspase 3 (Response Fig. 14A and Fig. S7J and lines 479-480 in the manuscript).

NAC treatment reduced body weights in $LysM^{Cre/+}$ $SerpinB2^{fl/fl}$ mice (Response Fig. 14B and Fig. S7G and lines 471-472 in the manuscript). Some of the beneficial effects of NAC could be attributed to the decreased body weights. We have now acknowledged this in Discussion (lines 584-585 in the manuscript).

6) The authors use a myeloid COX10 KO mouse to provide evidence that enhanced OXPHOS may be detrimental to ATMs and lead to increased ROS and death. However,

this data is only shown in supplemental and is missing the seahorse data to show the impact of COX10 KO on BMDM metabolism.

We have now performed the SeaHorse assay in BMDM isolated from WT and myeloid-specific *Cox10* KO mice. This experiment uncovered depressed basal and maximal respiration and spare capacity while extracellular acidification rates were unchanged in *Cox10*-deficient BMDM

compared to WT BMDM (Response Fig. 15 and Fig. S5D & S 5E and lines 370-372 in the manuscript).

Minor:

1) In Figure 1 N why did the authors use autofluorescence? Macrophage specific staining with either CD68 or F4/80 would be better here.

Figure 1N represents the intravital microscopy data of the dynamics of adipose monocyte-derived and resident macrophages at different timepoints after HFD initiation. We were unable to stain the macrophages *in vivo* with an antibody. However, we have performed flow cytometry in these mice to show that CCR2⁺ macrophages (F4/80⁺) express tdTomato (Response Fig. 11 and Fig. S1B and lines 143-147 in the manuscript).

2) As mentioned in major comments panel 1M should include both percentage and total number of macrophages as this will be hugely different between STD vs. HFD fed animals.

We have now included the total number of both macrophage subsets (Response Fig. 8A and Fig. S1I and lines 171-173 in the manuscript).

3) The authors should include the flow gating scheme with representative images for ATMs in the supplemental data.

The gating strategy for flow cytometry has been included in the supplemental data (Response Fig. 16 and Fig. 1G).

4) In figure 5 K they authors show Mitosox staining, which reveals subtle differences between WT and SerpinB2 KO macrophages. They then report a ratio of Mitosox to Mitotracker. Was this based on flow determined MFI values? Does this mean that mitochondrial number was also different between genotypes to drive the ratio up? Yes, these are flow cytometry data. We provided the ratios of mitosox to mitotracker to assess mitochondrial ROS content per unit of mitochondria. Although we cannot determine mitochondrial number, we can assess mitochondrial volume using

the mitotracker green dye by flow cytometry. We now show that mitotracker green MFI was significantly decreased in *Serp1b2*^{-/-} BMDM treated with palmitate (Response Fig. 17 and Fig. S4E and lines 342-344 in the manuscript).

5) The protocol for IL-4 reconstitution is missing from the methods. Did they use IL-4/IL-4R complexes or just naked cytokine?

Here is the protocol: Ten weeks old *LysM*^{cre/+}*Serp1b2*^{fl/fl} mice were fed with an HFD for two months followed by i.p infusion of 1 ng IL-4 (R&D, #404-ML-050/CF, 2 doses/ week) dissolved in 100 microliters of PBS (lines 600-603 in the manuscript).

6) In panel 5M the displayed MFI quantification is not congruent with the adjacent flow plot. The numbers for MFI must x 1000. This should be indicated.

We apologize for this oversight. We have now corrected the Y axis label of the graph.

References:

1. Kim, H.H. *et al.* xCT-mediated glutamate excretion in white adipocytes stimulates interferon-gamma production by natural killer cells in obesity. *Cell Rep* **42**, 112636 (2023).
2. Chawla, A., Nguyen, K.D. & Goh, Y.P. Macrophage-mediated inflammation in metabolic disease. *Nat Rev Immunol* **11**, 738-749 (2011).
3. Yao, J. *et al.* Macrophage IRX3 promotes diet-induced obesity and metabolic inflammation. *Nat Immunol* **22**, 1268-1279 (2021).
4. Lawler, H.M. *et al.* Adipose Tissue Hypoxia, Inflammation, and Fibrosis in Obese Insulin-Sensitive and Obese Insulin-Resistant Subjects. *J Clin Endocrinol Metab* **101**, 1422-1428 (2016).
5. Cox, N. *et al.* Diet-regulated production of PDGF α by macrophages controls energy storage. *Science* **373** (2021).
6. Hofwimmer, K. *et al.* IL-1 β promotes adipogenesis by directly targeting adipocyte precursors. *Nat Commun* **15**, 7957 (2024).
7. Choy, L. & Derynck, R. Transforming growth factor- β inhibits adipocyte differentiation by Smad3 interacting with CCAAT/enhancer-binding protein (C/EBP) and repressing C/EBP transactivation function. *J Biol Chem* **278**, 9609-9619 (2003).
8. Annevelink, C.E., Sapp, P.A., Petersen, K.S., Shearer, G.C. & Kris-Etherton, P.M. Diet-derived and diet-related endogenously produced palmitic acid: Effects on metabolic regulation and cardiovascular disease risk. *J Clin Lipidol* **17**, 577-586 (2023).

9. Qiu, T. *et al.* Obesity-induced elevated palmitic acid promotes inflammation and glucose metabolism disorders through GPRs/NF-kappaB/KLF7 pathway. *Nutr Diabetes* **12**, 23 (2022).
10. Vasamsetti, S.B. *et al.* Apoptosis of hematopoietic progenitor-derived adipose tissue-resident macrophages contributes to insulin resistance after myocardial infarction. *Sci Transl Med* **12** (2020).
11. Amit, I., Winter, D.R. & Jung, S. The role of the local environment and epigenetics in shaping macrophage identity and their effect on tissue homeostasis. *Nat Immunol* **17**, 18-25 (2016).
12. Lavin, Y. *et al.* Tissue-resident macrophage enhancer landscapes are shaped by the local microenvironment. *Cell* **159**, 1312-1326 (2014).
13. Yona, S. *et al.* Fate mapping reveals origins and dynamics of monocytes and tissue macrophages under homeostasis. *Immunity* **38**, 79-91 (2013).
14. Jaitin, D.A. *et al.* Lipid-Associated Macrophages Control Metabolic Homeostasis in a Trem2-Dependent Manner. *Cell* **178**, 686-698 e614 (2019).
15. Wang, X. *et al.* Prolonged hypernutrition impairs TREM2-dependent efferocytosis to license chronic liver inflammation and NASH development. *Immunity* **56**, 58-77 e11 (2023).

REVIEWERS' COMMENTS

Reviewer #1 (Remarks to the Author):

The authors have adequately responded to many of the suggestions. However, I do still have several remaining issues.

We appreciate the insightful comments from this reviewer. Addressing the concerns has helped us significantly improve the manuscript.

The manuscript still remains (too) information dense making it hard to read and digest all data that is presented. This also seems to go hand in hand with the fact that the manuscript basically presents two stories as one. The part on adipocyte size regulation seems a somewhat different story and is also not mentioned in the abstract. I would advise to reconsider this part again.

We have kept the adipocyte component in the manuscript in accordance with the previous suggestion of the editor. However, the reviewer has accurately pointed out that the abstract lacked the information on adipocyte size. We have now included this information in the abstract.

I also feel that the inclusion of the data obtained from human VAT samples is somewhat difficult to interpret. These samples were obtained from deceased people introducing many potential confounders. Also, age and BMI is hugely variable and no information on cause of death is presented.

We acknowledge these limitations of the patient samples and include this text in Discussion: “Another limitation of this study is that we used samples from deceased patients, which may introduce many potential confounders. These patients had highly variable BMI, and the information regarding their causes of death was not available”

Note, there is also no need to present BMI with 3 decimals. Also, in the M&M section the study participants are described as people with and without diabetes, while in the results section the participants are presented as lean vs. obese. Some of the study participants do have diabetes, yet based on their BMI would be classified as overweight. Also, how is diabetes being classified and is this type 1 or type 2 diabetes?

We have now provided the BMI with 2 decimals. These patients had type 2 diabetes. We have categorized these patients in either lean or obese based on their BMI. This information is now in Methods.

Related to the animal study with the LysM ^{+/+} Ifngr1 fl/fl and LysMcre^{-/+} Ifngr1 fl/fl mice, some results are difficult to interpret. Although total BW is similar, VAT adipose tissue is smaller.

What about total adipose tissue mass? Anything known about activity, food intake of the animals ? I guess multiple factors are at play here beyond the local effects of the macrophages in VAT.

These are all excellent points that reviewers have noted. In response to this comment, we have acknowledged the effects of these variables in Discussion: “Although our data indicate that myeloid-specific deficiency of *Infgr1* resulted in improved insulin sensitivity by protecting adipose-resident macrophages from apoptosis, we cannot rule out the contributions of reduced food intake, increased activity, and reduced fat mass to the beneficial effects.”.

Reviewer #2 (Remarks to the Author):

The authors have provided additional results and discussion, addressing each point raised in the first review, particularly the attributes used to define "residential" versus "infiltration" macrophages. I have no further questions for the current version.

We thank the reviewer for his constructive critiques, which helped us improve the manuscript significantly.

Reviewer #3 (Remarks to the Author):

In the revised manuscript the authors have largely addressed my concerns. The added significant new data and reorganized the manuscript, which significantly improves the impact of the study. I have no further comments.

We appreciate the positive feedback of this reviewer.